# Exploitable mechanisms of antibody and CAR mediated macrophage cytotoxicity

Tianyi Liu[1,2], Meng Zhang [1,2], Tatyanah Farsh [1,2], Haolong Li [1,2], Audrey Kishishita[3], Abhilash Barpanda[4], Stanley G. Leung[1,2], Jun Zhu[1,2], Hyuncheol Jung[5,6], Junjie Tony Hua[1,2], Xiaolin Zhu [1,6], Alexander B. Kim [7,8], Young Ah Goo[9], Minsoo Son [9], Jaenyeon Kim [9], Aish Subramanian[1,2], Martin Sjöström [2,10,11], Katherine C. Fuh [1,12], Jocelyn S. Chapman [1,12], Julia Carnevale [1,5,6,13], Luke A. Gilbert [1,14,15], Aparna Lakkaraju[16,17], Peter M. Bruno [1,14], David Quigley [1,14,18], Arun P. Wiita [4,19,20], Felix Y. Feng [1,2,14,21] & Carl J. DeSelm [7,8,21] ✉

Macrophages infiltrate solid tumors and either support survival or induce cancer cell death through phagocytosis or cytotoxicity. To uncover regulators of macrophage cytotoxicity towards cancer cells, we perform two co-culture CRISPR screens using CAR-macrophages targeting different tumor associated antigens. Both identify ATG9A as an important regulator of this cytotoxic activity. In vitro and in vivo, ATG9A depletion in cancer cells sensitizes them to macrophage-mediated killing. Proteomic and lipidomic analyses reveal that ATG9A deficiency impairs the cancer cell response to macrophage-induced plasma membrane damage through defective lysosomal exocytosis, reduced ceramide production, and disrupted caveolar endocytosis. Depleting non-cytotoxic macrophages using CSF1R inhibition while preventing ATG9A-mediated tumor membrane repair enhances the anti-tumor activity of therapeutic antibodies in mice. Thus, macrophage cytotoxicity plays an important role in tumor elimination during antibody or CAR-macrophage treatment, and inhibiting tumor membrane repair via ATG9A, particularly in combination with cytotoxic macrophage enrichment through CSF1R inhibition, improves tumor-targeting macrophage efficacy.

Tumor-associated macrophages (TAMs) are pivotal components of the tumor microenvironment, either promoting or combating cancer progression directly or indirectly through T-cells. T-cells are generally recognized as the main immune mediators of tumor cell death, and tumor mutational burden (TMB) is now a well-recognized predictor of response to many current immune therapies. Direct antitumor effects of macrophages are achieved by phagocytosis or tumor cytotoxicity, and unlike T-cells, do not rely on tumor neoantigens, which makes them particularly well suited to overcome tumors with low TMB. However, the direct killing capacity of a macrophages is low. Numerous approaches attempt to enhance the phagocytic capacity of macrophages, such as by combining antitumor antibodies to engage pro-phagocytic pathways in macrophages with blockade of "don't eat me" receptors[1,2], however, the effect of these approaches have been surprisingly ineffective at eliminating tumor in patients[3].

Recent advances to further improve macrophage tumor-specific phagocytosis leverages a Chimeric Antigen Receptor (CAR), which also utilizes a tumor-binding antibody, directly expressed on the macrophage surface as an extracellular domain with a chimeric transmembrane and intracellular pro-phagocytic signal[4,5]. We generated CARs with the same intracellular signaling domain as used by antibodies – the Fc Receptor, which signals through ITAMs. The

dominant mechanism by which macrophages directly kill cancer cells, especially when directed to target them with an antibody or a CAR, remains unestablished; better understanding of this mechanism may lead to more effective macrophage directed therapies. Macrophages are often thought to phagocytose and degrade live cancer cells upon recognizing tumors via antibody or CAR binding, a concept influenced by their well-established role in clearing necroptotic dead cells[6]. To test whether direct live tumor phagocytosis is central to tumor killing by macrophages, and to further elucidate the pathways critical to CAR-directed macrophage killing of cancer cells in an unbiased manner, we performed parallel CRISPR screens using two different CARs targeting two different tumor antigens expressed by a human solid cancer cell line.

While many co-culture screens have been conducted to study the interaction between cancer cells and immune cells, such as T cells[7,8] or NK cells[9,10], relatively few have focused on macrophages, and most of these studies target liquid tumors[11]. Macrophages are particularly abundant in solid tumors, including those classified as immune-cold[12]. Immune-cold tumors are characterized by low neoantigen expression and immune cell infiltration, resulting in a less effective T-cell mediated immune response against the tumor[13]. Macrophages in these environments play a critical dual role. They can suppress any existing T-cell response through the secretion of inhibitory cytokines. However, they also have the potential to amplify T-cell activity or directly exert anti-tumor effects if properly activated.

We use ovarian cancer as a tumor model for these studies, because not only is it a solid tumor with overall poor outcomes, but it is also relatively immunologically cold, with a large tumor associated macrophage component, and poor response to numerous forms of immune therapy[14].

While we identified previously known "don't eat me" receptors such as CD47 and APMAP[2,11] in our CAR-M co-culture CRISPR screens, the vast majority of key molecules and pathways identified to be important to tumor death by macrophages were not connected to phagocytosis, suggesting that other mechanisms may be involved in tumor directed macrophage killing. ATG9A, a member of the autophagy pathway, was one major hit without a clear connection to tumor phagocytosis or cytotoxicity. Autophagy, a highly conserved cellular process, plays a crucial role in maintaining cellular homeostasis by degrading and recycling damaged organelles and proteins[15]. ATG9A acts as a membrane carrier, shuttling between various intracellular compartments and the forming autophagosome membrane, thus contributing significantly to the regulation of membrane trafficking during autophagy[16]. Beyond its canonical role, ATG9A has been implicated in lipid mobilization, immune regulation, and plasma membrane repair[7,17,18].

To better define how tumor-targeting macrophages eliminate cancer cells, dual co-culture CRISPR screens in ovarian cancer models using distinct CAR-macrophages both identify ATG9A as a regulator whose loss increases tumor death after direct macrophage contact, independent of phagocytosis. Functional studies show that ATG9A facilitates repair of membrane damage, which is induced by cytotoxic macrophages. We find that that combining strategies to 1) modulate cytotoxic macrophage composition in the tumor microenvironment (e.g., CSF1R inhibition), 2) enhance macrophage targeting (e.g., antibody or CAR-based approaches), and 3) impair tumor membrane repair (e.g., ATG9A depletion) together can achieve complete tumor responses independent of T-cells.

## Results

### Co-culture CRISPR screens nominated surface regulators of CAR-macrophage-mediated cytotoxicity

We performed two co-cultured CRISPR screens in the ovarian cancer cell line OVCAR-8, using two distinct CAR-THP-1 cell lines: liquid tumor relevant CD19-targeting CAR-THP-1 (Supplementary Fig. 1A-B), and solid tumor relevant EphA2-targeting CAR-THP-1 (Supplementary Fig. 1C-D). Unless otherwise specified, these CAR-THP-1 cells will be referred to as α-EphA2-Ms and α -CD19-CAR-Ms, respectively throughout this paper. Both CARs contain the antigen-specific scFv followed by an extracellular linker, transmembrane domain, and human FcGRIIA-derived intracellular domain; this is the same domain engaged by antibody binding to Fc receptors on macrophages.

To assess CAR-macrophage cytotoxicity, we used EphA2-expressing T-24 and OVCAR-8 cells, along with CD19-expressing Raji cells and CD19+ OVCAR-8 cells. Flow cytometry confirmed target antigen expression (Supplementary Fig. 1E-H). Live-cell imaging co-culture experiments evaluated targeted killing efficacy. Cancer cells were NucLight Red-labeled and co-cultured with either α-CD19-CAR-Ms or α-EphA2-CAR-Ms, compared to untransduced (UTD) macrophages. CAR-Ms showed significantly higher killing efficiency, confirming their specificity for CD19+ and EphA2+ cancer cells (Supplementary Fig. 1I-L).

To identify targetable genes crucial for the survival of cancer cells in the presence of tumor-targeting macrophages, we developed a CRISPR sublibrary targeting 1,378 highly expressed cell surface-associated genes across various cancer types (Fig. 1A, Supplementary Data 1). We performed CRISPR screens using parental OVCAR-8 cells that endogenously express EphA2 and CD19+ OVCAR-8 cells, followed by 1-week co-culture with or without CAR-Ms. sgRNA abundance was then analyzed using next-generation sequencing and the MAGeCK pipeline (Fig. 1C-D).

Differential scores are plotted in Fig. 1E, F (Supplementary Data 2–3). A positive score suggests gene knockout (KO) confers increased resistance in cancer cells to macrophage-induced cytotoxicity, while a negative score indicates increased susceptibility. EphA2, which is endogenously expressed by OVCAR-8 tumor, emerged as a top resistance hit in the α-EphA2-CAR-M screen, confirming the validity of our method, as target loss prevents CAR-M recognition. CD19 did not rank highly in the CD19 overexpressed CAR-M screen, likely due to the discrepancy between the overexpressed codon-optimized CD19 sequence and the endogenous sequence recognized by the guide RNA.

We selected the top 10 hits with negative scores from both screens, primarily focusing on non-essential genes. Individual sgRNA knockouts of the top 10 hits were performed in CD19+ OVCAR-8 cells, which were then co-cultured with parental macrophages, α-EphA2-CAR-Ms, or α-CD19-CAR-Ms. Most guide KOs did not affect cell proliferation, except for UBAP1 (Ubiquitin Associated Protein 1) KO (Supplementary Fig. 2A). ATG9A KO and UBAP1 KO demonstrated significant sensitizing effects, consistent with their identification as top hits from the screen (Fig. 2A). Importantly, this sensitization effect was not CAR-specific, as indicated by a significant sensitization score in co-cultures with parental macrophages. UBAP1, a component of the ESCRT complex, is crucial for membrane remodeling and protein sorting, processes important for cancer cell plasma membrane repair following T cell-induced damage[19]. Given the role of the ESCRT machinery, it is plausible that UBAP1 and other ESCRT factors identified in the screen (VPS25 and CHMP6), confer resistance to macrophages via a similar mechanism. ATG9A (Autophagy Related 9 A) is a key protein involved in the autophagy process. Although its role in macrophage sensitization was unclear, it has been shown that ATG9A regulates cancer cell response to T-cell-mediated killing through the IFN-γ pathway[7].

ITGAV (Integrin Alpha V) was identified as a common resistant hit from both screens. Increased resistance to macrophage-mediated killing was observed in sgITGAV CD19+ OVCAR-8 cells (Supplementary Fig. 2B). In co-culture experiments using SKOV-3 cells and α-EphA2-CAR-Ms, UBAP1 KO and ATG9A KO sensitized the cells to macrophage cytotoxicity, whereas ITGAV KO conferred resistance (Supplementary

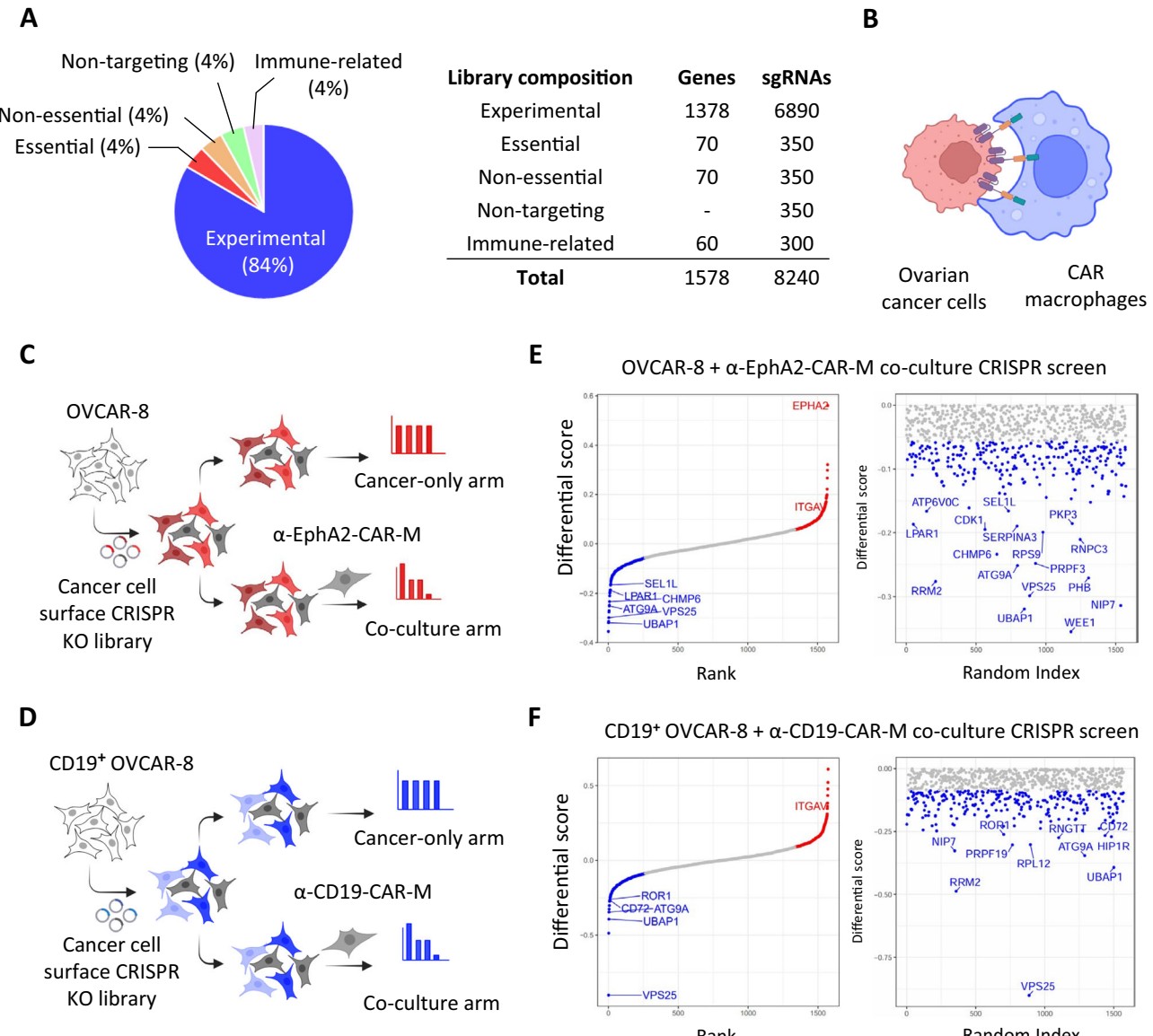

**Fig. 1 | Co-culture CRISPR screens nominated surface regulators to CAR-macrophage-mediated cytotoxicity. A** Composition of the cell surface-associated CRISPR knockout library, including experimental, essential, non-essential, immune-related, and non-targeting guides. **B** Schematic showing CAR-M attacking an ovarian cancer cell. **C-D** Overview of CRISPR co-culture screens using OVCAR-8 cells and either α-EphA2-CAR-Ms (**C**) or α-CD19-CAR-Ms (**D**), with cancer-only and co-culture arms.

**E, F** Screen results showing ranked gene-level differential scores for α-EphA2-CAR-M (**E**) and α-CD19-CAR-M (**F**) screens, highlighting candidate genes that modulate macrophage-mediated killing. Some elements of this figure were created with BioRender.com and are included under a publication license in accordance with BioRender's user agreement. *Created in BioRender. DeSelm, C. (2025)* https://BioRender.com/q45rp4i; *BioRender. DeSelm, C. (2025)* https://BioRender.com/b4cxqxy.

---

Fig. 2C). These consistent results across different CARs and cell lines highlight the robustness of our screens.

Based on IncuCyte phenotype, we narrowed down to five hits with the most significant sensitization and aimed to further validate these findings through a competition assay. NucLight Green (NL GREEN) OVCAR-8 cells infected with sgGAL4 (hereafter referred to as control), or NucLight Red (NL RED) cells with test sgRNAs, were mixed at one-to-one ratio and co-cultured with macrophages for 6 days (Supplementary Fig. 2D). We observed a significant reduction in ATG9A KO and UBAP1 KO cancer cells compared to control cancer cells when cocultured with macrophages (Fig. 2B, C). Considering that our initial screen was conducted using a THP-1 macrophage cell line, we sought to validate these findings leveraging primary human peripheral blood mononuclear cell (PBMC)-derived macrophages. In this context, ATG9A KO demonstrated the most significant reduction in cell survival compared to other guides (Supplementary Fig. 2E). ATG9A KO also

showed the most substantial sensitizing effect in competition assays using SKOV-3 cells and THP-1 macrophages (Supplementary Fig. 2F). KO efficiency of individual guides was shown in Supplementary Fig. 2G, H.

Given ATG9A's strong phenotype, we tested its functional role by reintroducing ATG9A into ATG9A KO cells, which rescued resistance to α-EphA2-CAR-Ms, confirming its specific role in macrophage-induced cytotoxicity (Supplementary Fig. 2I). Flow cytometry analysis showed that ATG9A KO did not alter the expression of key phagocytosis checkpoints (MHC-I, PD-L1, CD24, CD47) or the CAR target EphA2 (Supplementary Fig. 2J), suggesting ATG9A regulates cancer cell susceptibility to macrophage killing through a specific and independent mechanism.

To confirm ATG9A's clinical relevance, we analyzed TCGA data[20] and found that lower ATG9A mRNA levels correlate with improved overall survival in ovarian serous cystadenocarcinoma (Supplementary

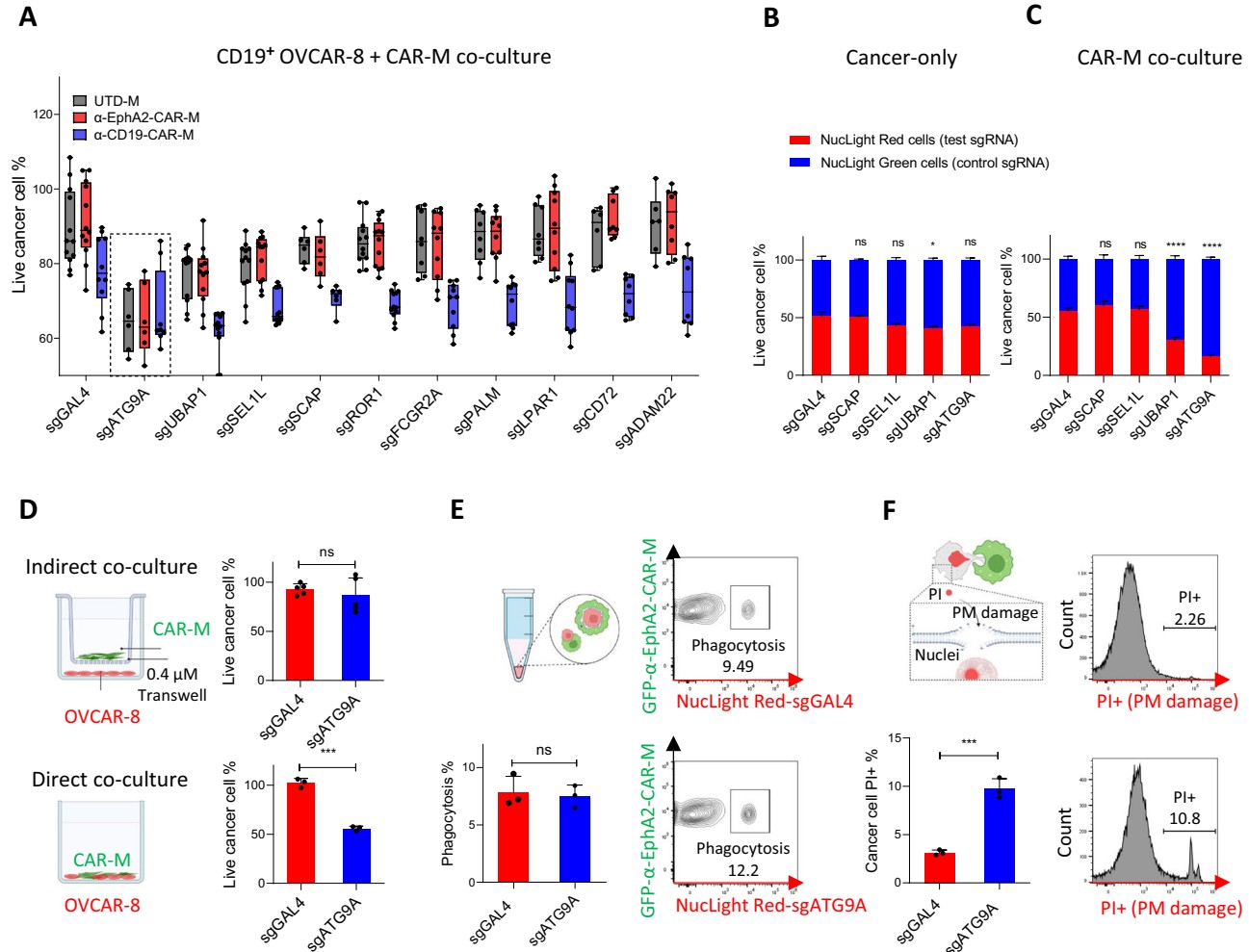

**Fig. 2 | ATG9A protects ovarian cancer cells from CAR-M-induced plasma membrane damage. A** Individual guides KO co-culture experiment to validate top hits of the screen. CD19⁺ OVCAR-8 cells were cultured alone or with UTD, α-EphA2-CAR or α-CD19-CAR macrophages for 3 days. Live cancer cell % was calculated by normalizing cell counts in the co-cultured group by the cancer-cell-only group at day three (n = 6 biological replicates; mean ± SEM). Box plots showed the median (center line), 25th–75th percentiles (box), and 2.5th–97.5th percentiles (whiskers), with all data plotted. **B, C** Nuclight Green (NL Green) labeled-OVCAR-8 cells were infected with control guides (sgGAL4); Nuclight Red (NL Red) labeled-cells were infected with test guides. Green cells and red cells were mixed at a one-to-one ratio and were cultured alone or with macrophages for 6 days. n = 3 biological replicates; mean ± SEM. One-way ANOVA followed by Dunnett's multiple comparisons test was used to determine statistical significance. **D** sgGAL4 and sgATG9A OVCAR-8

cells were co-cultured with α-EphA2-CAR-Ms in regular 12-well plate or in a 12-well plate with 0.4 μM Transwell inserts for 3 days. (n = 3 biological replicates; mean ± SEM. Statistical significance was determined using two-tailed unpaired Student's t-tests. **E** NL Red-labeled sgGAL4 or sgATG9A OVCAR-8 cells and GFP-labeled α-EphA2-CAR-Ms were co-cultured for 2 h. Flow cytometry was used to quantify the percentage of double-positive cells out of total GFP+ macrophages (n = 3 biological replicates; mean ± SEM. Statistical significance was determined using two-tailed unpaired Student's t-tests. **F** PI (propidium iodide) PM integrity assay after 2 h co-culture. (n = 3 biological replicates; mean ± SEM. Statistical significance was determined using two-tailed unpaired Student's t-tests. *P < 0.05; **P < 0.01; ***P < 0.001. Some elements of this figure were created with BioRender.com and are included under a publication license in accordance with BioRender's user agreement. *Created in BioRender. DeSelm, C. (2025)* https://BioRender.com/hv2dnxb.

Fig. 3A). Among the ten top hits, ATG9A had the highest pan-cancer Z-score and the second highest Z-score in ovarian cancer, strongly linking it to poor prognosis (Supplementary Fig. 3B). We also found lower ATG9A mRNA is associated with better survival in several additional cancer types (Supplementary Fig. 4).

In conclusion, dual CRISPR screens identified ATG9A and UBAP1 as crucial regulators of cancer cell susceptibility to CAR-macrophage-induced cytotoxicity. Knockouts of these genes increase cancer cell sensitivity to CAR-macrophage killing, justifying further studies to elucidate the mechanism.

#### ATG9A protects ovarian cancer cells from CAR-M-induced plasma membrane damage

Given its promising phenotype scores, strong link to poor cancer prognosis, as well as relatively unexplored relevance to macrophage

response, we focused on ATG9A in our following study. ATG9A, the sole transmembrane protein in the core autophagy gene family, plays a key role in the autophagy pathway, regulating autophagosome formation crucial for cellular material degradation and survival[16]. Autophagy is critical for cancer cell survival and resistance to diverse forms of cell death, including immune-mediated killing[21].

Macrophages eliminate cancer cells through various mechanisms, including phagocytosis and secretion or expression of cytotoxic molecules[12]. Recent research also suggests that macrophages may kill cancer cells by nibbling on their cell membranes[5]. To determine whether ATG9A KO-induced sensitization depends on direct cancer-macrophage contact, we conducted co-culture experiments with a 0.4 μM pore insert segregating α-EphA2-CAR-Ms on top. We showed that direct contact is necessary for the sensitizing effect of ATG9A KO (Fig. 2D). Therefore, we focused on two major killing mechanisms

requiring cell-cell contact: phagocytosis and plasma membrane damage.

To evaluate whole-cell phagocytosis, we co-cultured NL RED-labeled OVCAR-8 control or ATG9A KO cells with GFP-labeled α-EphA2-CAR-Ms for 2 h and quantified the double-positive percentage after gating out doublets. This assay requires macrophages to engulf entire cancer cells, including their nuclei, to become double positive. No difference in phagocytosis rates was observed between macrophages co-cultured with control or ATG9A KO cancer cells. (Fig. 2E). The same result was observed in a PH-based live cell microscopy phagocytosis assay (Supplementary Fig. 5A, B). To assess cancer cell membrane integrity after macrophage co-culture, we used Propidium Iodide (PI), which stains nuclei if the plasma membrane is compromised. Significantly more PI-positive cells were observed in ATG9A KO compared to control shortly after co-culture (Fig. 2F), indicating increased cancer cell plasma membrane damage induced by macrophages in the absence of ATG9A.

We repeated the flow-based phagocytosis and PI integrity assays using primary PBMC-derived macrophages, confirming consistent phenotypes (Supplementary Fig. 5C). To validate this effect with a different CAR construct, we generated HER2-CAR macrophages, which similarly showed enhanced cancer cell sensitization after ATG9A KO (Supplementary Fig. 5D). A 3D co-culture model further confirmed the sensitizing effect of ATG9A KO (Supplementary Fig. 5E). Notably, macrophage growth rates, measured by GFP mean fluorescence intensity (MFI), remained comparable between groups (Supplementary Fig. 5F).

Antibody-dependent macrophage cytotoxicity is a killing mechanism where macrophages recognize antibody-coated target cells via Fc receptors and release cytotoxic mediators to induce cell death[22–24]. To evaluate the effect of ATG9A knockout in this context, we utilized trastuzumab, a HER2-targeted therapeutic antibody, to activate macrophage-mediated killing. In vitro, Trastuzumab alone resulted in moderate tumor cell killing in the absence of macrophages; however, in a macrophage co-culture setting, trastuzumab significantly enhanced tumor cell killing in sgGAL4, and particularly sgATG9A co-cultures (Supplementary Fig. 5G).

To test the significance of this effect in vivo, we engrafted sgGAL4 and sgATG9A SKOV-3 cells into NOD.Cg-Prkdc scid Il2rg tm1Wjl/SzJ (NSG) mice, which lack all immune cells except monocytes and macrophages, and treated them with Trastuzumab to enhance the antibody-dependent macrophage cytotoxicity (Fig. 3A). Over 6 weeks, we observed a robust reduction in the growth of ATG9A KO tumors as compared to control tumors when treated with Trastuzumab, demonstrating that ATG9A ablation sensitizes SKOV-3 cancer cells to macrophage-mediated killing induced by Trastuzumab; no growth difference was observed in human IgG isotype group, indicating that ATG9A KO did not impact baseline cell proliferation or survival in vivo (Fig. 3B).

To confirm macrophage involvement, we repeated this experiment with the addition of clodronate liposomes to deplete the endogenous macrophages (Fig. 3C). Flow cytometry staining for F4/80 confirmed effective macrophage depletion by clodronate (Supplementary Fig. 5H). As expected, macrophage depletion with clodronate blunted the therapeutic effect of ATG9A KO. Notably, no growth differences were observed between sgGAL4 and sgATG9A tumors under trastuzumab treatment when clodronate was used to deplete the endogenous macrophages (Fig. 3D).

To reconstitute and further assess CAR-macrophage-mediated killing in vivo, we pretreated NSG mice with clodronate to deplete endogenous macrophages before injecting intraperitoneal SKOV-3 tumors. Once tumors were established, we treated sgGAL4 and sgATG9A tumors with five million α-EphA2-CAR-Ms (Fig. 3E). Tumor growth was monitored over 4 weeks using luciferase imaging. α-EphA2-CAR-Ms demonstrated significantly greater efficacy in controlling sgATG9A tumors relative to control tumors (Fig. 3F).

Our study demonstrates that ATG9A knockout sensitizes cancer cells to macrophage-mediated killing through mechanisms requiring direct cell-cell contact, involving direct membrane damage rather than whole cell phagocytosis. In vivo experiments using subcutaneous and intraperitoneal SKOV-3 models further confirmed that ATG9A ablation enhances macrophage-mediated killing of cancer cells, suggesting that targeting ATG9A may improve responses to macrophage-based therapies.

## ATG9A regulation of plasma membrane dynamics impairs cancer response to damage

Previous findings suggest a potential role for ATG9A in the plasma membrane repair process triggered by microbial and endogenous agents[18]. ATG9A is a key membrane protein in autophagy, primarily localized to intracellular membranes such as the endoplasmic reticulum, Golgi apparatus, and autophagosomes. Interestingly, we observed an increased localization of ATG9A-RFP at the plasma membrane in co-culture with macrophages (Fig. 4A, B). This suggests that ATG9A translocated to the plasma membrane in response to macrophage attack and may play a role in facilitating the membrane repair process.

Calcium is crucial for membrane repair, as it triggers lysosomal exocytosis and vesicle fusion at injury sites[25]. To determine whether ATG9A-mediated repair is $Ca^{2+}$-dependent, we conducted co-culture experiments with or without $Ca^{2+}$. In its presence, ATG9A KO OVCAR-8 cells showed increased CAR-M-induced membrane damage, while in its absence, both control and KO cells exhibited similar PI+ levels, indicating impaired repair (Fig. 4C). Notably, $Ca^{2+}$ itself did not affect CAR-M's ability to induce damage (Supplementary Fig. 6A). These findings suggest ATG9A facilitates $Ca^{2+}$-dependent membrane repair in response to macrophage-induced injury, rather than simply reducing membrane susceptibility. To monitor early-stage membrane disruptions that may not permit PI uptake, a HaloTag probe is utilized alongside fluorescent ligands that are either membrane-permeant (MPL) or membrane-impermeant (MIL) (Fig. 4D). Quantifying the percentage of MIL+ cells among all HaloTag-GFP + OVCAR-8 cells in CAR-M co-culture revealed significantly more membrane damage in sgATG9A cells when exposed to CAR-Ms (Fig. 4E).

Macrophages can induce plasma membrane damage on cancer cells through cytotoxic factors like reactive oxygen species (ROS) and reactive nitrogen species (RNS), which potentially induce lipid peroxidation in damaged membranes[26]. To define macrophage-induced plasma membrane changes at the lipid level, membrane samples were isolated from flow-sorted pre- and post-co-culture cancer cells and subjected to lipidomic or proteomic analysis (Fig. 4F). We identified lipid features associated with hydroxylation, epoxylation, and hydroperoxylation, processes typically mediated by ROS. Nitration, primarily driven by nitric dioxide radicals, was not detected. Significantly more peroxidized lipids were found in ATG9A KO cells after macrophage co-culture compared to control cells (Fig. 4G). This finding was validated using a lipid peroxidation sensor, showing a significant increase of oxidized/reduced lipids ratio in ATG9A KO cells post-macrophage co-culture (Fig. 4H). Rescue experiments using NO scavengers (L-NMMA and Carboxy-PTIO) and ROS scavengers (Mito-TEMPOL and Apocynin) revealed that ROS inhibition exhibited a more significant rescuing effect in ATG9A KO cells (Fig. 4I).

ATG9A has been reported to protect the plasma membrane from damage caused by microbial and endogenous agents through collaboration with IQGAP1 and the ESCRT system[18].We also confirmed enhanced ATG9A-IQGAP1 interaction in co-cultured cancer cell samples compared to cancer-only conditions (Supplementary Fig. 6B). However, IQGAP1 knockout in OVCAR-8 cells did not sensitize them to CAR-Ms killing, suggesting IQGAP1 is not essential for this process (Supplementary Fig. 6C).

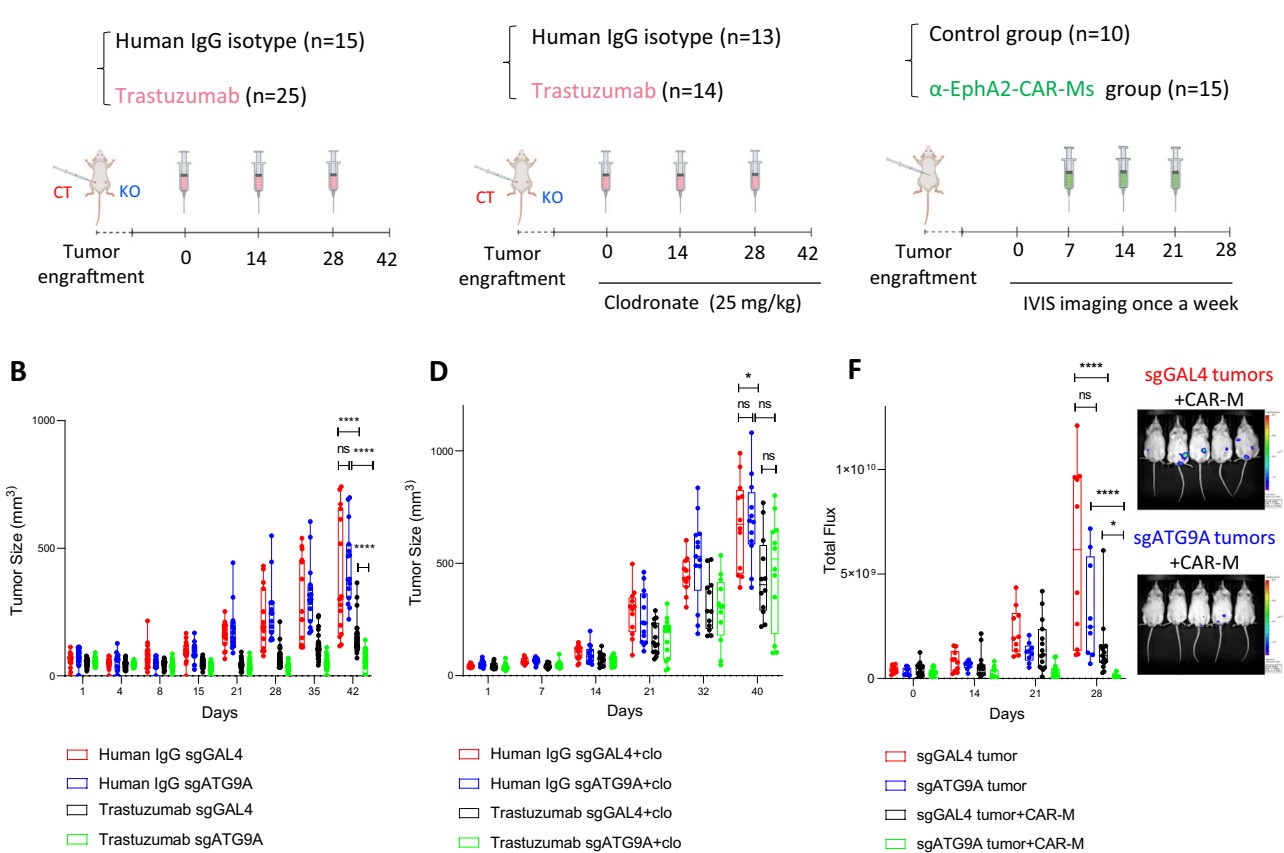

**Fig. 3 | ATG9A ablation enhances macrophage-mediated tumor control in vivo.**
**A, B** NSG Mice bearing control (CT) or ATG9A knockout (KO) subcutaneous SKOV3 tumors were treated with IgG ($n = 15$), or Trastuzumab ($n = 25$). Tumor volume was monitored over time. Box plots show the median (center line), 25th–75th percentiles (box), and 2.5th–97.5th percentiles (whiskers), with all data plotted. Statistical significance was determined using two-way analysis of variance (ANOVA) followed by Tukey's post-hoc test. **C, D** NSG Mice bearing control or KO subcutaneous SKOV3 tumors were treated with IgG ($n = 13$), or Trastuzumab ($n = 14$) in combination with clodronate (25 mg/kg) to deplete macrophages. Box plots show the median (center line), 25th–75th percentiles (box), and 2.5th–97.5th percentiles (whiskers), with all data plotted. Statistical significance was determined using two-way analysis of variance (ANOVA) followed by Tukey's post-hoc test. **E, F** Schematic

and results of an intraperitoneal (IP) SKOV3-luciferase tumor model treated with or without α-EphA2-CAR macrophages (CAR-Ms). Tumor burden was quantified via bioluminescence imaging once weekly ($n = 10$ for control, $n = 15$ for CAR-M group). Box plots show the median (center line), 25th–75th percentiles (box), and 2.5th–97.5th percentiles (whiskers), with all data plotted. Representative IVIS images are shown on the right. Statistical significance was determined using two-way analysis of variance (ANOVA) followed by Tukey's post-hoc test. *$P < 0.05$; **$P < 0.01$; ***$P < 0.001$. Some elements of this figure were created with BioRender.com and are included under a publication license in accordance with BioRender's user agreement. *Created in BioRender. DeSelm, C. (2025)* https://BioRender.com/ddmwmhn.

To further investigate whether the sensitization induced by ATG9A knockout is ESCRT-dependent, we further KO ATG9A in sgUBAP1 and sgCHMP6 OVCAR-8 cells (Supplementary Fig. 6D). IncuCyte killing assay showed enhanced sensitivity to α-EphA2-CAR-M killing in ESCRT KO cells after ATG9A KO (Supplementary Fig. 6E). This suggests that the role of ATG9A in regulating cancer cell response to macrophages is not ESCRT-dependent.

Given the apparent importance of ATG9A at the plasma membrane, we conducted surface mass spectrometry proteomics to explore changes in cell surface proteins after ATG9A knockout in OVCAR-8 cells (Fig. 5A). Surface proteomics analysis revealed widespread downregulation of pathways related to cell adhesion, cell junctions, and membrane rafts following ATG9A knockout, indicating a disruption in cellular architecture.

Interestingly, we observed significant downregulation of three caveolae proteins: caveolae associated protein 2 (CAVIN2), caveolae associated protein 1 (CAVIN1) and caveolin-1 (CAV1), with CAVIN2 identified as a top hit. Western blot confirmed the downregulation of CAVIN1 and CAVIN2 following ATG9A knockout (Fig. 5B). Caveolae play crucial roles in plasma membrane repair by facilitating calcium influx,

organizing signaling molecules, and maintaining lipid homeostasis[27]. Live cell imaging revealed increased colocalization of ATG9A-RFP and CAVIN2-GFP in cancer cells co-cultured with macrophages (Fig. 5C), a finding further supported by co-immunoprecipitation experiments (Supplementary Fig. 6B). Importantly, knocking out CAVIN1 or CAVIN2 in OVCAR-8 cells phenocopied the effects of ATG9A knockout (Fig. 5D, E), suggesting both ATG9A and caveolae components are critical for maintaining plasma membrane integrity under macrophage attack. CAVIN1 and CAVIN2 KO efficiency was confirmed by Western blot (Supplementary Fig. 6F).

In our surface proteomics analysis, we also observed an increase in calreticulin levels, a phagocytosis checkpoint, following ATG9A knockout. Despite these changes (Supplementary Fig. 6G), calreticulin knockdown did not affect the cancer cell response to CAR-Ms in either control or ATG9A KO OVCAR-8 cells, indicating these are likely secondary changes (Supplementary Fig. 6H).

These findings suggest that ATG9A collaborates with caveolae components to maintain plasma membrane integrity and that its absence sensitizes cancer cells to macrophage-mediated killing through the inability to efficiently repair ROS-induced lipid

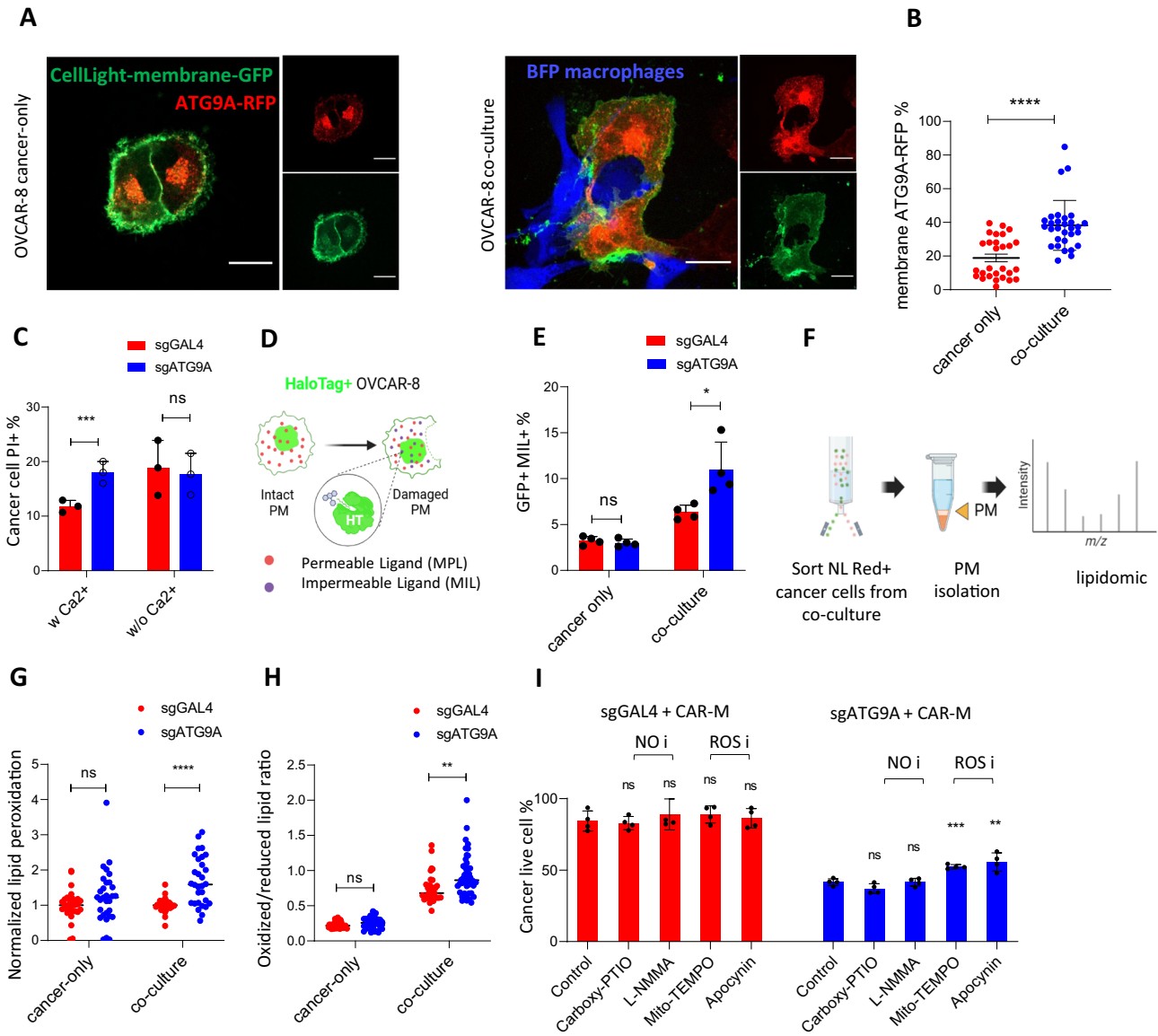

**Fig. 4 | ATG9A protects ovarian cancer cells from CAR-macrophage-induced membrane damage and lipid peroxidation. A** Confocal images showing OVCAR-8 cells infected with ATG9A-RFP plasmid and Myr-Palm-GFP plasmid co-cultured alone or with BFP-labeled macrophages. Scale bar = 20 μm. **B** Quantification of ATG9A-RFP + /Myr-Palm-GFP+ area normalized by ATG9A-RFP+ area in OVCAR-8 cells (*n* = 30 images, mean ± SEM). Statistical significance was determined using two-tailed unpaired Student's *t*-tests. **C** Control or sgATG9A OVCAR-8 cells were cultured with α-EphA2-CAR-macrophages for 2 h with or without Ca²⁺ (*n* = 3 biological replicates; mean ± SEM). Statistical significance was determined using two-way ANOVA followed by Tukey's multiple comparisons test. **D**, **E** HaloTag-labeled OVCAR-8 cells were co-cultured with α-EphA2-CAR-macrophages for 2 h with permeable or impermeable ligands (*n* = 4 biological replicates; mean ± SEM). Statistical significance was determined using two-way ANOVA followed by Tukey's multiple comparisons test. **F** Schematic showing membrane samples isolated from flow-sorted cancer cells were submitted for lipidomics. **G** Normalized lipid peroxidation was calculated by dividing the abundance of peroxidized lipids by the abundance of lipids prior to transformation (*n* = 26 lipid species; mean ± SEM). Statistical significance was determined using two-way ANOVA followed by Tukey's multiple comparisons test. **H** Lipid peroxidation in control or KO OVCAR-8 cells pre- or post-co-culture was quantified by a lipid peroxidation sensor (*n* = 35 biological replicates; mean ± SEM). Statistical significance was determined using two-way ANOVA followed by Tukey's multiple comparisons test. **I** IncuCyte co-culture experiment with different inhibitors blocking NO or ROS (*n* = 4 biological replicates; mean ± SEM). Statistical significance was determined using two-way ANOVA followed by Tukey's multiple comparisons test. *$P < 0.05$; **$P < 0.01$; ***$P < 0.001$. Some elements of this figure were created with BioRender.com and are included under a publication license in accordance with BioRender's user agreement. *Created in BioRender. DeSelm, C. (2025)* https://BioRender.com/7iu7qmq.

peroxidation. We next examined exactly how cancer cells utilize ATG9A to facilitate repair of the plasma membrane.

## ATG9A facilitates cancer cell membrane repair after macrophage-induced damage by recruiting ceramide to sites of damage

To examine whether lipid-level modifications shed more insight into the precise mechanism by which ATG9A facilitates repair of ROS-induced membrane damage by macrophages, we compared all lipid species from control or ATG9A KO cells identified in our membrane lipidomic data (Supplementary Fig. 7A, Supplementary Data 4). Significant ceramide increase was found in control cell plasma membrane post-macrophage co-culture, but this difference was not observed in ATG9A KO samples, indicating disrupted ceramide localization to the plasma membrane upon ATG9A knockout (Fig. 6A).

Sphingomyelinase is crucial for converting sphingomyelin into ceramide on the plasma membrane. Neutral sphingomyelinase (nSMase) primarily localizes at the plasma membrane, whereas acid

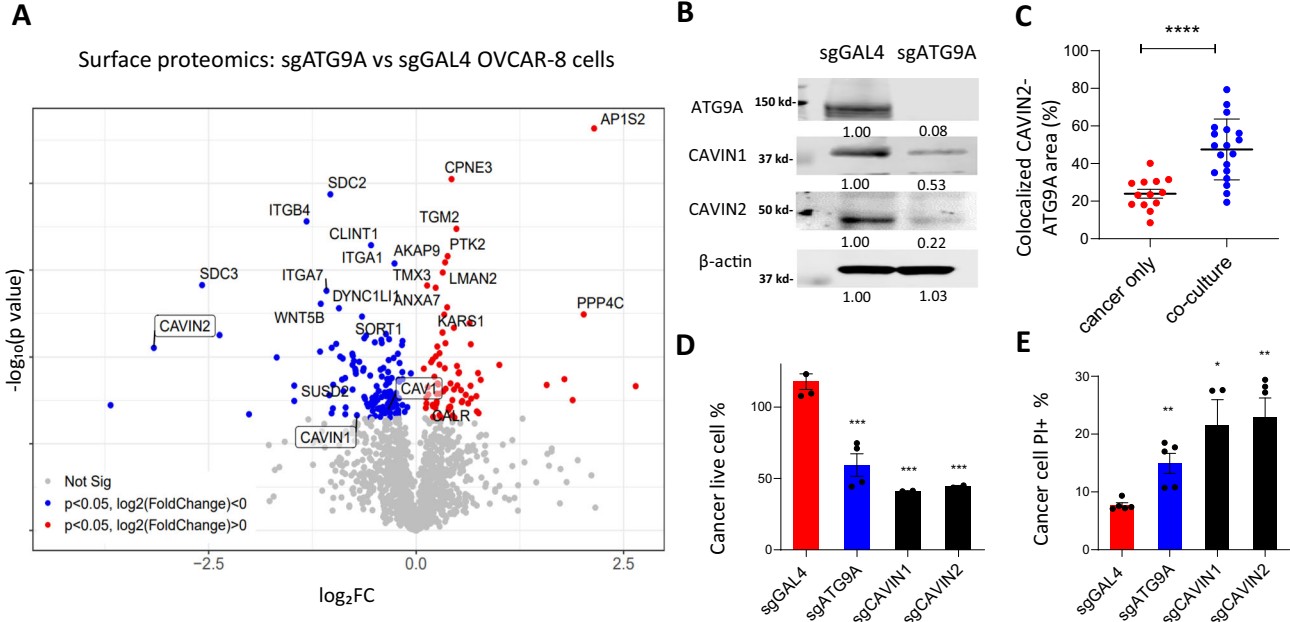

**Fig. 5 | ATG9A loss impairs caveolae structure and sensitizes ovarian cancer cells to CAR-macrophage cytotoxicity. A** Surface proteomic profiling comparing control (sgGAL4) and ATG9A knockout (sgATG9A) OVCAR-8 cells revealed significant downregulation of multiple caveolae-associated proteins, including CAVIN1 and CAVIN2 (highlighted with boxes). **B** Western blot validation confirming reduced expression of CAVIN1 and CAVIN2 upon ATG9A loss. **C** Quantification of CAVIN2-ATG9A colocalization in OVCAR-8 cells transfected with ATG9A-RFP and CAVIN2-GFP, either cultured alone or co-cultured with CAR macrophages for 24 h ($n = 15$ biological replicates, mean ± SEM). Statistical significance was determined using two-tailed unpaired Student's $t$-tests. **D** Three-day IncuCyte co-culture experiment using control, ATG9A KO, CAVIN2 KO and CAVIN2 KO OVCAR-8 cells and α-EphA2-CAR macrophages ($n = 3$ biological replicates; mean ± SEM). Statistical significance was determined using two-tailed unpaired Student's $t$-tests. **E** PI uptake assay measuring plasma membrane permeability OVCAR-8 cells infected with different sgRNAs co-cultured with α-EphA2-CAR macrophages ($n = 5$ biological replicates; mean ± SEM). Statistical significance was determined using two-tailed unpaired Student's $t$-tests. $*P < 0.05$; $**P < 0.01$; $***P < 0.001$.

sphingomyelinase (ASMase) is typically lysosomal but can translocate to the plasma membrane via lysosome exocytosis in response to stress or damage[28]. Ceramides are critical in plasma membrane repair as they facilitate membrane curvature and caveolae formation necessary for resealing damaged areas[20]. Finally, caveolar endocytosis helps in the removal and recycling of damaged membrane components[29] (Fig. 6B).

Our immunofluorescence staining data confirmed an increase in membrane ceramide in control cells post-co-culture, but not in ATG9A KO cells (Fig. 6C, D). Similarly, the surface LAMP1 level significantly increased in control cells after co-culture, indicative of lysosome exocytosis; in contrast, the surface LAMP1 level remained unchanged in ATG9A KO cells throughout the co-culture experiment (Fig. 6E). This suggests defective lysosomal exocytosis in ATG9A KO cells, leading to disrupted ceramide regulation in response to macrophage interactions, implicating ATG9A in plasma membrane repair of macrophage induced damage.

Both amitriptyline and GW4869 inhibit ceramide production but target different enzymes. Amitriptyline inhibits ASMase, while GW4869 inhibits nSMase[30,31]. Inhibiting ASMase pharmacologically eliminated the differences in CAR-M-induced killing and plasma membrane damage between control and ATG9A KO OVCAR-8 cells. In contrast, inhibiting nSMase by GW4869 had no effect on these differences (Fig. 6F, G). Furthermore, we confirmed that recombinant ASMase reverses the sensitizing effect of ATG9A KO in OVCAR-8 cells to CAR-M-mediated killing and plasma membrane damage (Fig. 6H, I).

In summary, our lipidomic analysis revealed disrupted lysosome as well as ceramide localization to the plasma membrane due to ATG9A knockout. This disruption impairs plasma membrane repair and can be rescued by providing exogenous ASMase.

To expand on our surface proteomics and lipidomics findings, we performed whole-cell proteomics to assess the broader impact of ATG9A KO (Supplementary Fig. 7B, C). After 3 days of co-culture with

α-EphA2-CAR-Ms, proteomic analysis revealed upregulated IFN-γ and TNF-α signaling, corroborating previous reports on autophagy inhibition and inflammatory responses[15]. Interestingly, pathways associated with fatty acid metabolism and oxidative phosphorylation were downregulated. Prior research indicates that ATG9A facilitates lipid transfer from droplets to mitochondria, supporting oxidative phosphorylation[17]. Given the critical roles of lipids in membrane repair, we wonder whether ATG9A is involved in lipid transport necessary for plasma membrane restoration.

Lipid droplet analysis showed increased lipid droplet size and number in ATG9A KO cells, indicating disrupted lipid metabolism (Supplementary Fig. 7D, E). Additionally, LAMP1 colocalization with lipid droplets increased in control cells post-co-culture, indicating macrophage-induced lipid droplet delivery to lysosomes, a response absent in ATG9A KO cells (Supplementary Fig. 7F, G).

These findings demonstrate that ATG9A is essential for maintaining cellular integrity under macrophage-induced stress. ATG9A facilitates lipid transport from lipid droplets to lysosomes, and lysosomes to damaged plasma membrane. Disruption of these processes in ATG9A-deficient cells reveals potential vulnerabilities in membrane repair pathways that may be exploited for cancer therapy.

## ATG9A-mediated plasma membrane repair after macrophage-induced damage is autophagy-independent

ATG9A, the only core autophagy gene included in our cell surface library due to its unique role as a transmembrane autophagy protein, requires further investigation to determine if its function in plasma membrane repair is autophagy-dependent.

We first assessed the impact of ATG9A knockout on the autophagy pathway. Western blot analysis revealed a significant increase in LC3A/B-II expression in control OVCAR-8 cells after inhibiting autophagosome degradation by chloroquine, suggesting baseline

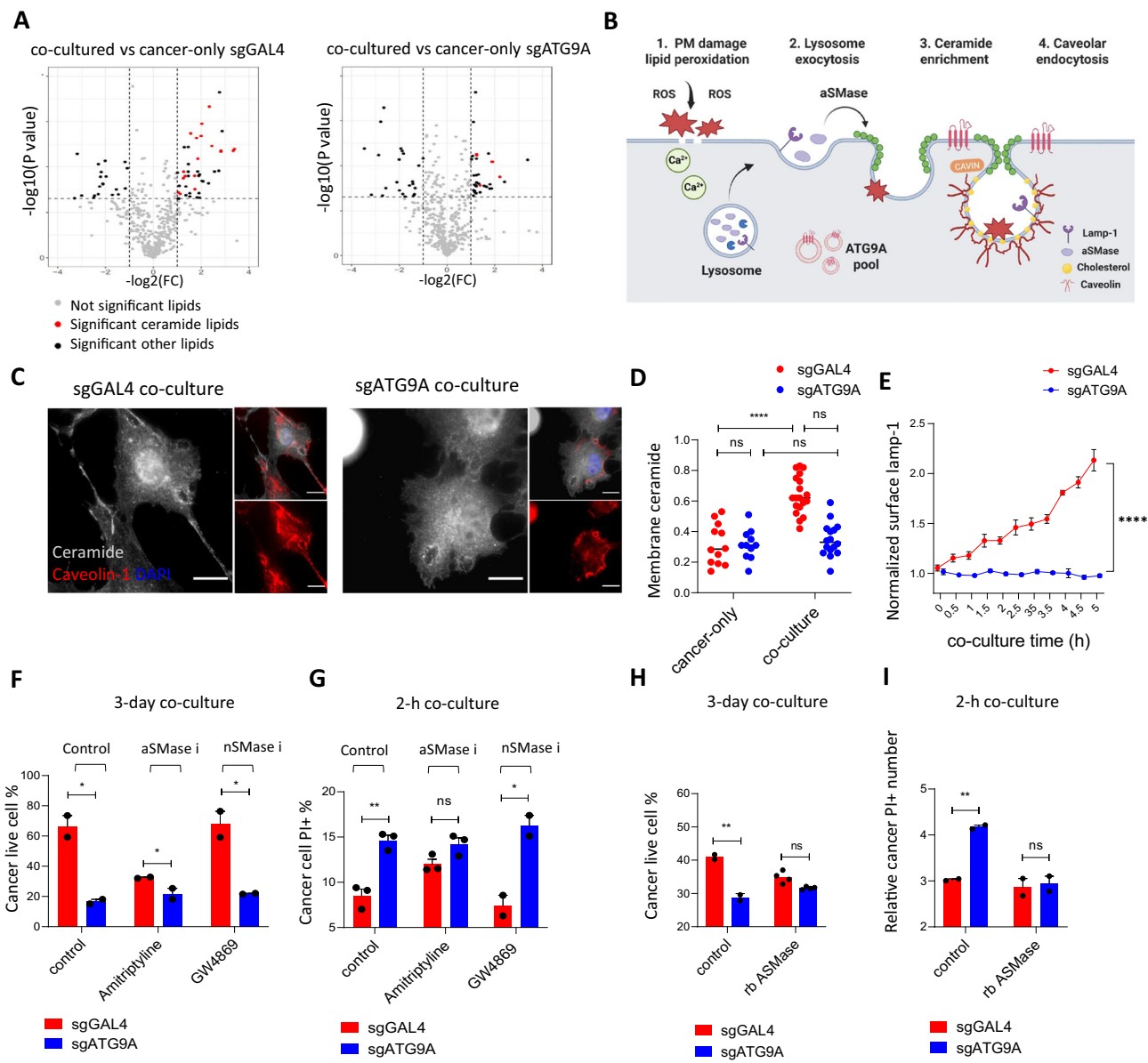

**Fig. 6 | ATG9A facilitates cancer cell membrane repair after macrophage-induced damage by recruiting ceramide to sites of damage. A** Lipidomics was performed using isolated membrane samples from control and KO OVCAR-8 cells from cancer-only group or sorted from α-EphA2-CAR macrophage co-culture (*n* = 2 biological replicates). Ceramide species were highlighted in red. **B** Schematic demonstrating the plasma membrane repair process on cancer cell surface upon macrophage attack. **C-D** Immunofluorescent staining was performed using anti-ceramide and anti-caveolin-1 antibodies. Scale bar = 20 μm. Ceramide colocalization with caveolin-1 was quantified as Pearson correlation coefficient (*n* = 12 biological replicates, mean ± SEM). Statistical significance was determined using two-way ANOVA with Tukey's multiple comparisons test. **E** Control and KO OVCAR-8 cells were co-cultured with α-EphA2-CAR macrophages or cultured alone; Cancer surface lamp-1 expression was quantified by flow cytometry (*n* = 4 biological replicates; mean ± SEM). Statistical significance was determined using two-tailed unpaired Student's *t*-tests. **F** Live cell percentage was calculated by the cell count of

co-cultured OVCAR-8 cells treated with media, Amitriptyline, or GW4869 normalized by their cancer-only controls (*n* = 3 biological replicates; mean ± SEM). Statistical significance was determined using two-way ANOVA with Tukey's multiple comparisons test. **G** PM integrity assay was performed using control or KO OVCAR-8 cells cultured with α-EphA2-CAR macrophages for 2 h under different treatments (*n* = 3 biological replicates; mean ± SEM). Statistical significance was determined using two-way ANOVA with Tukey's multiple comparisons test. **H, I** sgGAL4 or sgATG9A OVCAR-8 cells were co-cultured with α-EphA2-CAR macrophages with or without recombinant (rb) ASMase (*n* = 3 biological replicates; mean ± SEM). Statistical significance was determined using two-way ANOVA with Tukey's multiple comparisons test. **P* < 0.05; ***P* < 0.01; ****P* < 0.001. Some elements of this figure were created with BioRender.com and are included under a publication license in accordance with BioRender's user agreement. *Created in BioRender. DeSelm, C. (2025)* https://BioRender.com/5ap5yy9.

autophagic flux. Autophagic flux was impaired following ATG9A knockout, suggested by less increase in LC3A/B-II expression (Fig. 7A). Additionally, significantly elevated p62 expression levels were observed after ATG9A knockout in both OVCAR-8 and SKOV-3 cells, confirming impaired autophagy in the absence of ATG9A (Fig. 7B).

As summarized in Fig. 7C, ATG9A is integral to autophagosome formation in the autophagy process and aids in lipid breakdown within

autolysosomes[17,32]. It is involved in membrane trafficking between the Golgi apparatus, trans-Golgi network (TGN), and endosomes, cycling between these compartments[33,34]. Additionally, we showed that ATG9A plays a critical role in plasma membrane repair by regulating endocytosis and lysosomal exocytosis.

Autophinib, a small molecule inhibitor of autophagy[35], was utilized in the competition assay to investigate whether

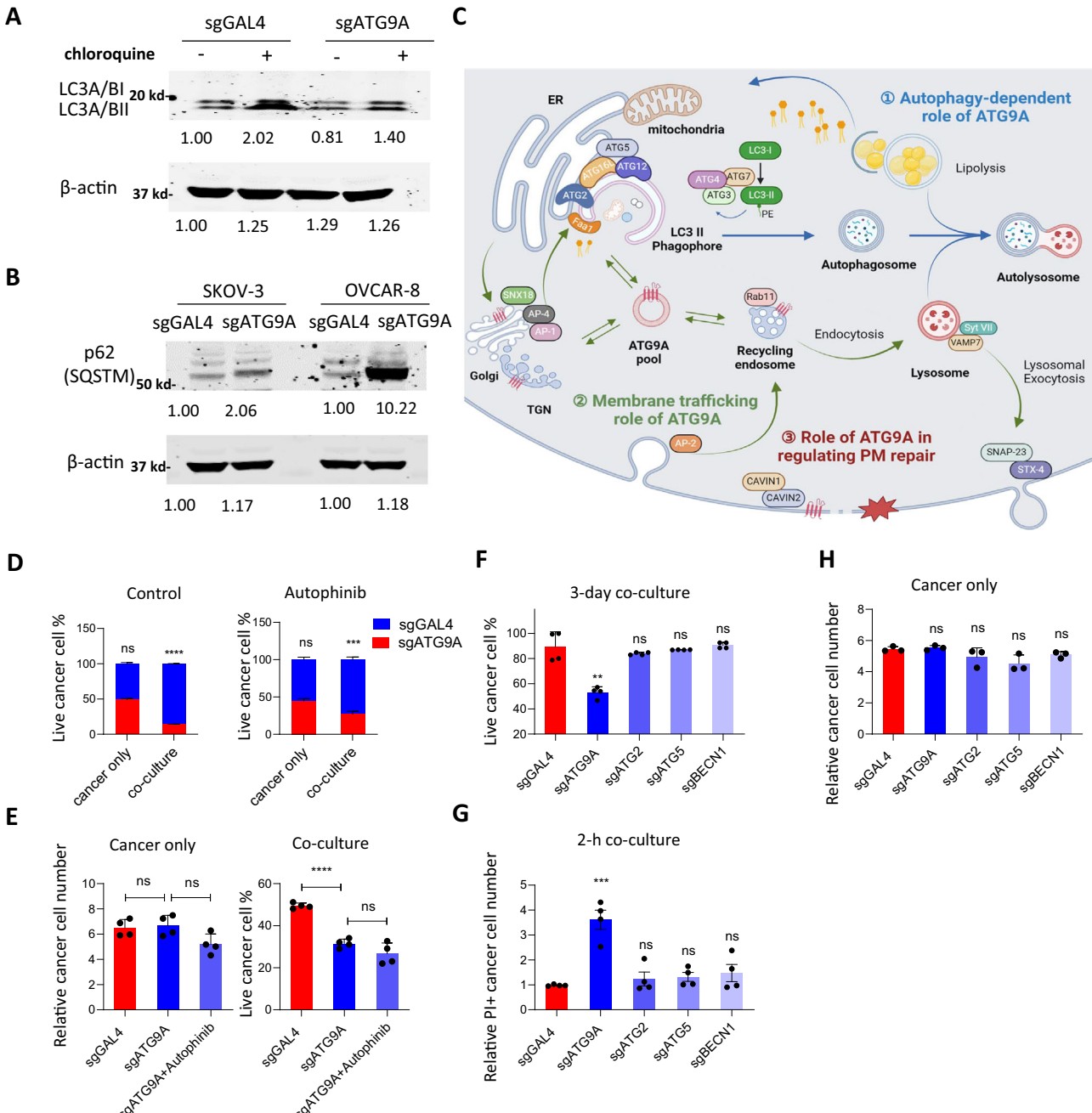

**Fig. 7 | ATG9A-mediated plasma membrane repair after macrophage-induced damage is autophagy-independent.** **A** Western blot analysis of autophagic flux in control and KO OVCAR-8 cells. Chloroquine was added to prevent autophagosome degradation. **B** Western blot using control or sgATG9A SKOV-3/OVCAR-8 cells to examine p62 expression. **C** Overview of different roles of ATG9A. LC3: Microtubule-associated proteins 1 A/1B light chain 3; PE: Phosphatidylethanolamine; SNX18: Sorting Nexin 18; AP-1: Adapter Protein Complex 1; AP-2: Adapter Protein Complex 2; Rab11: Ras-related protein Rab-11A; VAMP7: Vesicle-associated membrane protein 7; Syt VII: Synaptotagmin-7; SNAP-23: Synaptosomal-associated protein, 23 kDa; STX-4: Syntaxin-4; TGN: Trans-Golgi Network; Faa1: Fatty acid amide hydrolase 1; ER: Endoplasmic Reticulum. **D** Competition assay using regular media or media with Autophinib ($n = 3$ biological replicates; mean ± SEM). Statistical significance was determined using two-

tailed unpaired Student's t-tests. **E** CAR-M Co-culture experiments using sgGAL4, sgATG9A OVCAR-8 cells and sgATG9A cells with Autophinib treatment ($n = 4$ biological replicates; mean ± SEM). Statistical significance was determined using two-way ANOVA with Tukey's multiple comparisons test. **F-G** Live cell percentage or relative PI+ number were normalized to the cell count or PI+ cell number in the cancer-only group ($n = 4$ biological replicates; mean ± SEM). Statistical significance was determined using two-tailed unpaired Student's t-tests. **H** Cell proliferation at day 3 normalized to cell number at day 0 ($n = 3$ biological replicates; mean ± SEM). Statistical significance was determined using two-tailed unpaired Student's t-tests. *$P < 0.05$; **$P < 0.01$; ***$P < 0.001$. Some elements of this figure were created with BioRender.com and are included under a publication license in accordance with BioRender's user agreement. *Created in BioRender. DeSelm, C. (2025)* https://BioRender.com/xrev1ob.

autophagy inhibition mitigates ATG9A KO-induced sensitization. After a six-day co-culture, ATG9A KO cells remained more sensitive than controls, even with autophagy inhibition (Fig. 7D). Similarly, in sgATG9A OVCAR-8 co-cultures with CAR-Ms, the level of CAR-M killing was unchanged with or without Autophinib,

indicating that the addition of Autophinib did not enhance sensitization (Fig. 7E).

To test the importance of canonical autophagy more definitively, we investigated if knocking out other core autophagy genes (ATG2, ATG5, BECN1) produces a similar sensitization effect to CAR-M

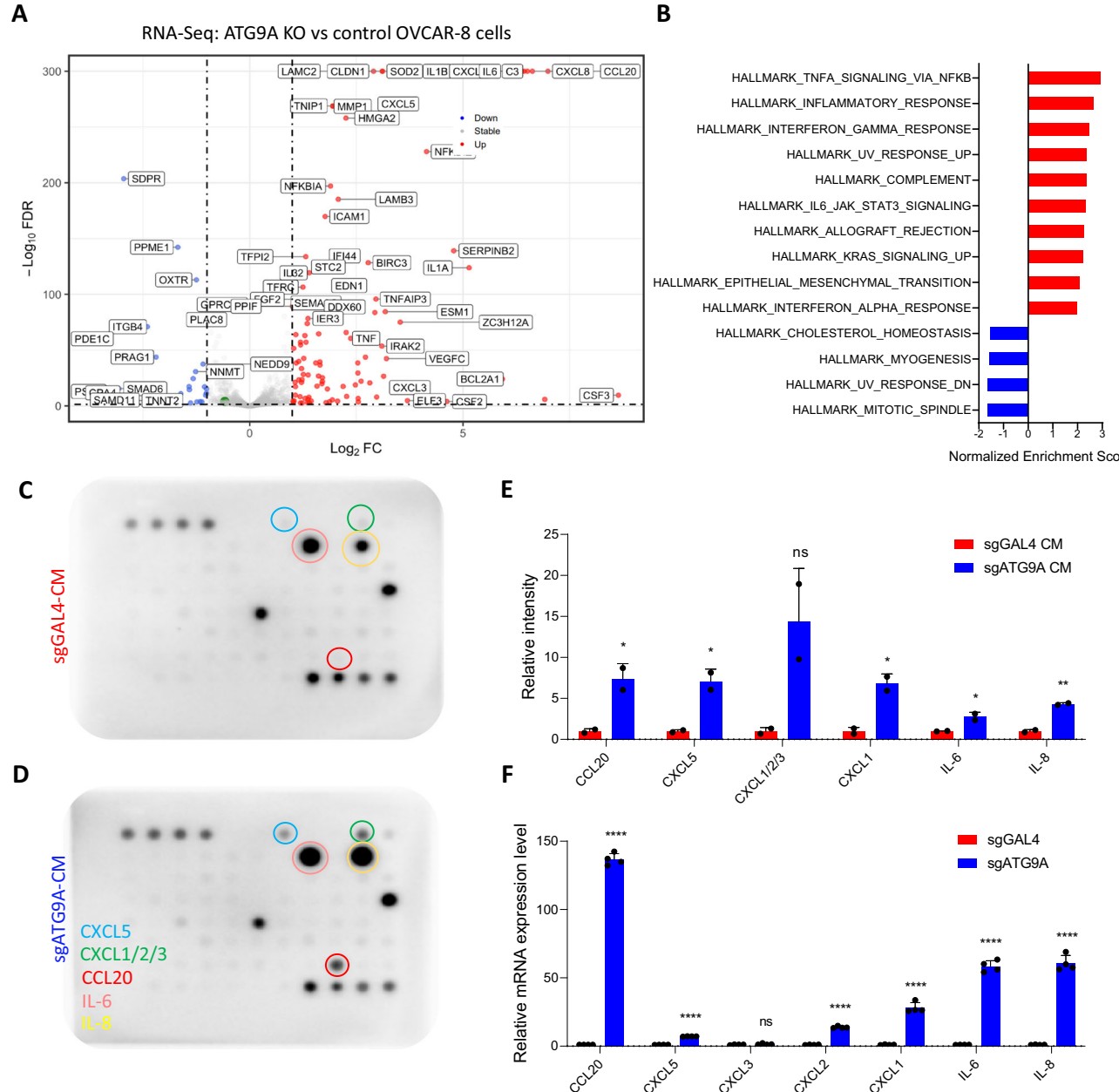

**Fig. 8 | ATG9A KO ovarian cancer cells induced inflammatory cytokine secretion in vitro. A** RNA-seq analysis using control and ATG9A KO OVCAR-8 cells. **B** Gene Set Enrichment Analysis (GSEA) using the Hallmark gene set. **C–E** Cytokine array using conditioned media derived from control or ATG9A KO OVCAR-8 cells ($n = 2$ biological replicates; mean ± SEM). Statistical significance was determined using unpaired two-sided $t$-tests for each cytokine. **F** qRT-PCR confirming upregulation of inflammatory cytokines in sgATG9A OVCAR-8 cells compared to control cells. ($n = 4$ biological replicates; mean ± SEM). Statistical significance was determined using unpaired two-sided $t$-tests for each cytokine. *$P < 0.05$; **$P < 0.01$; ***$P < 0.001$.

cytotoxicity. After co-culture with CAR-Ms, only ATG9A KO showed increased membrane damage, and similarly only ATG9A KO resulted in increased cancer cell killing (Fig. 7F, G). None of the knockouts affected cancer cell survival in the absence of macrophages (Fig. 7H). This highlights a unique, autophagy independent role for ATG9A in regulating cancer cell response to, and repair of macrophage-induced plasma membrane damage.

It is well-established that autophagy inhibition leads to inflammatory cytokine secretion[36,37]. RNA-Seq analysis of ATG9A KO cells revealed upregulated inflammatory cytokines, with GSEA showing activation of TNF-α and IFN-γ pathways (Fig. 8A, B). Cytokine array (Fig. 8C–E) and qRT-PCR (Fig. 8F) confirmed increased expression of IL-6, IL-8, CXCL1/3/5, and CCL20 in ATG9A KO cells. These findings suggest that ATG9A KO not only disrupts membrane repair after cytotoxic macrophage

exposure but also promotes an inflammatory cytokine environment, potentially enhancing macrophage activation and cytotoxicity.

Our study highlights the crucial role of ATG9A in sensitizing cancer cells to macrophage-induced membrane damage and killing, which is independent of autophagy. While we determined that macrophages induce tumor membrane peroxidation that cannot be efficiently repaired in the absence of ATG9A, we next sought to determine the precise nature of the macrophages responsible for this type of damage, in vivo.

## Combination treatment with Trastuzumab and CSF1R inhibitor induces tumor regression in ATG9A KO tumors

Given that ATG9A deficiency in tumor leads to increased inflammatory cytokine secretion, we aimed to characterize the tumor associated and

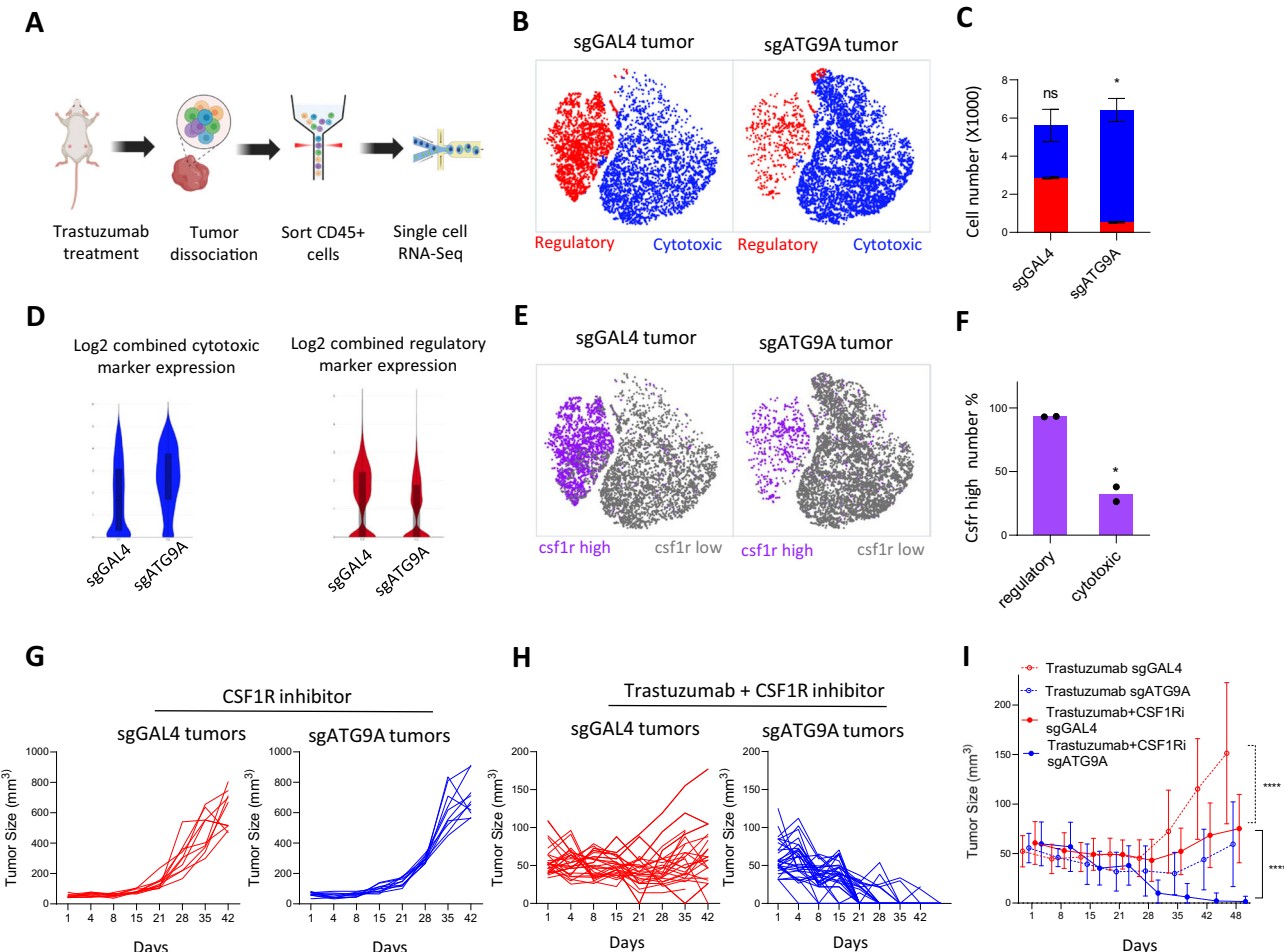

**Fig. 9 | Combination treatment with Trastuzumab and CSF1R inhibitor induces tumor regression in ATG9A KO tumors. A** Trastuzumab-treated sgGAL4 and sgATG9A tumors were processed for single-cell RNA-Seq library preparation (6 tumors from individual sgGAL4 mice were pooled into two samples [three tumors per sample], and the same was done for 6 tumors from sgATG9A mice. 10,000 cells were analyzed per sample). **B** t-distributed stochastic neighbor embedding (tSNE) plot shows clustering of cells (K-Means = 2). Two major clusters of macrophages were identified. Cytotoxic macrophages, marked by elevated *Nos, Cox2, Lcn2,* and *S100a8* expression; and regulatory macrophages, indicated by increased *Cx3cr1, Cd206, Tgf-β,* and *Il10* expression. **C** Cell numbers of cytotoxic and regulatory macrophages were quantified for both sgGAL4 and sgATG9A tumors (mean ± SEM). Statistical significance was determined using two-tailed unpaired Student's *t*-tests. **D** Violin plots show Log2 combined cytotoxic or regulatory macrophage markers expression. Violin plots display the distribution of gene expression across cells (via kernel density estimation), with an overlaid box plot showing the median and interquartile range (25th–75th percentiles); whiskers extend to the full data range (minimum and maximum values).

**E** Cells with high *csf1r* expression were plotted in purple, low-*csf1r*-cells were shown in gray. **F** *Csf1r* high percentage in the regulatory and cytotoxic clusters from ATG9A KO tumors were quantified (mean ± SEM). Statistical significance was determined using two-tailed unpaired Student's *t*-tests. scRNA-seq data was obtained from two biological replicates per condition; each replicate consisted of 10,000 cells pooled from 3 mice. While each cell provides an individual measurement, statistical comparisons between conditions should be interpreted with caution due to the limited number of pooled biological replicates (*n* = 2). **G, H** Tumor size summary of control or KO tumors under CSF1Ri alone (*n* = 10 tumors) or combined Trastuzumab + CSF1Ri treatment during 6 weeks (*n* = 30 tumors). **I** Summarized sgGAL4 and sgATG9A tumor growth under Trastuzumab alone (*n* = 25 tumors; mean ± SEM) or Trastuzumab + CSF1Ri (*n* = 30 tumors; mean ± SEM). Statistical significance was determined using two-way analysis of variance (ANOVA) followed by Tukey's post-hoc test. **P* < 0.05; ***P* < 0.01; ****P* < 0.001. Some elements of this figure were created with BioRender.com and are included under a publication license in accordance with BioRender's user agreement. *Created in BioRender. DeSelm, C. (2025)* https://BioRender.com/qqmbp7u.

cytotoxic macrophages in control and ATG9A KO tumors in vivo, which may have mechanistic and therapeutic relevance. We sorted CD45+ immune cells from Trastuzumab-treated control and ATG9A KO tumors in NSG mice and performed single-cell RNA sequencing (Fig. 9A). Gating strategy is shown in Supplementary Fig. 9. Nearly all cells exhibited high CD14 expression, confirming the immune population in NSG mice is almost exclusively composed of monocyte/macrophages, while lack of Ncr1 expression confirmed no meaningful NK cell population (Supplementary Fig. 8A, B).

First, we sought to understand whether the inflammatory environment induced by ATG9A KO cells modulates macrophage infiltration and polarization within tumors. After filtering out all CD14(-) stromal cells, T-distributed stochastic neighbor embedding (tSNE) plots revealed two main macrophage clusters.

Given ATG9A's role in membrane repair, we focused on cytotoxic macrophage markers associated with lipid damage and ROS production, including *Inos, Cox-2, S100a8,* and *Lcn2*[38,39] (Supplementary Fig. 8C). Additionally, we analyzed regulatory/anti-inflammatory markers, such as *Cx3cr1, Mrc1, Trem2,* and *Tgfb1* (Supplementary Fig. 8D).

Based on macrophage marker enrichment, we classified the two clusters as regulatory macrophages (high regulatory markers) and cytotoxic macrophages (high membrane-damaging proteins). In ATG9A KO tumors, regulatory macrophages decreased by 81.4%, while cytotoxic macrophages increased by 114.0% compared to controls (Fig. 9B, C). Violin plots confirmed this shift, showing higher cytotoxic and lower regulatory signatures in ATG9A KO tumors (Fig. 9D). These findings suggest that ATG9A KO-mediated sensitization to macrophage attack may involve not only impaired membrane repair after

exposure to cytotoxic macrophages, but also the induction of cytotoxic macrophages in vivo.

To further define their roles in vivo, we identified CSF1R as a distinguishing actionable marker of regulatory macrophages, with low expression in cytotoxic macrophages (Fig. 9E). In ATG9A KO tumors, most regulatory macrophages showed high CSF1R expression, whereas fewer cytotoxic macrophages expressed CSF1R (Fig. 9F). This suggests that CSF1R inhibition could preferentially deplete regulatory macrophages, shifting the balance toward cytotoxic macrophages. To test this, we combined Trastuzumab with the CSF1R inhibitor PLX5622 in NSG mice bearing SKOV3 tumors.

Remarkably, while CSF1R inhibition alone had no effect on sgGAL4 or sgATG9A tumor size (Fig. 9G), 93.3% (28/30) of mice bearing ATG9A KO tumors were completely eradicated when combined with Trastuzumab and CSF1Ri, compared to 0% in the control group (Fig. 9H, I). This highlights that depleting regulatory macrophages via CSF1R inhibition enhances cytotoxic macrophage activity, leading to robust tumor eradication when ATG9A-dependent membrane repair is impaired.

Staining with the pan-macrophage marker F4/80 revealed significantly more macrophages in Trastuzumab-treated ATG9A KO versus control tumors (Fig. 10A). CSF1R inhibitor reduced overall macrophage content to a significantly greater degree in control tumors than in ATG9A KO tumors, and their composition was skewed. CSF1R inhibition significantly reduced CX3CR1+ macrophages (which also expressed high CSF1R), but not *Inos*+ macrophages (which expresses low CSF1R), in both control and ATG9A KO tumors (Fig. 10B, C).

Our study demonstrates that ATG9A deficiency sensitizes cancer cells to macrophage-mediated killing, particularly in the presence of antitumor antibodies or CAR, which provide target specificity for cytotoxic macrophages. The combination of Trastuzumab and CSF1R inhibition to deplete non-cytotoxic macrophages significantly enhances ATG9A KO cancer cell killing in vivo. These findings support the potential of targeting ATG9A to improve the efficacy of macrophage-based cancer immunotherapies.

## Discussion

Our understanding of macrophage-mediated cytotoxicity in cancer is evolving. While antibody dependent cellular cytotoxicity (ADCC) is well described for NK cells and is heavily mediated by granzyme and perforin release[40], macrophages lack these cytotoxic molecules and appear to rely on alternative pathways for tumor cell elimination. Moreover, the dual nature of macrophages—capable of both promoting and inhibiting tumor growth depending on their polarization and the tumor microenvironment—adds significant complexity to their role in cancer biology. Clinical data on macrophage-mediated cancer cell killing is limited, necessitating further research to elucidate the conditions under which macrophages can effectively target and eliminate cancer cells.

In an effort to address these gaps, we performed a robust co-culture CRISPR screen in solid cancer cells to identify key cancer cell surface regulators of macrophage cytotoxicity. Our screen identified ATG9A along with several ESCRT genes (UBAP1, CHMP6, and VPS25) as critical determinants of the cancer cell response to macrophages. Functional knockouts of these genes revealed divergent impacts on macrophage-mediated cytotoxicity, implicating distinct underlying pathways. In particular, while ESCRT components—previously implicated in plasma membrane repair during T-cell responses[19]—suggest a shared mechanism of membrane repair exploited by macrophages, ATG9A appears to operate independently of ESCRT pathways.

Specifically, ATG9A KO cells exhibited increased sensitivity to macrophage-induced plasma membrane damage, independent of whole-cell phagocytosis. Loss of ATG9A significantly alters lipid metabolism and surface protein expression. Specifically, lipidomic data indicate that macrophage co-culture triggers ROS-mediated lipid peroxidation in ATG9A-deficient tumor cells, increasing membrane vulnerability. Moreover, ATG9A KO cells fail to induce lysosome exocytosis, thereby preventing ceramide-dependent plasma membrane repair. These findings emphasize ATG9A's essential role in orchestrating ceramide-mediated tumor repair mechanisms in response to macrophage-induced stress.

In addition, previous studies have also linked ATG9A to lipid metabolism, specifically in enabling lipid mobilization from lipid droplets[17]. It has also been shown that ATG9A-positive vesicles are directed toward the leading edge for exocytosis, facilitating the delivery of lipids and cargo proteins necessary for cell expansion[41]. Lipids are fundamental for plasma membrane repair. We observed downregulation of fatty acid metabolism and increased lipid droplet accumulation in ATG9A KO cells, as well as lipid droplet colocalization with lysosomes in control, but not ATG9A KO cells, after coculture with macrophages. This indicates ATG9A mediates lipid droplet delivery to lysosomes and lysosome delivery to the plasma membrane in response to tumor-targeting macrophage damage.

Furthermore, while autophagy is a critical process for cancer cell survival and has been highlighted in numerous CRISPR screens as a key regulator of immune responses, our studies indicate that ATG9A's role in macrophage-mediated membrane repair operates independently of canonical autophagy. RNA-Seq analysis revealed significant upregulation of pro-inflammatory cytokines, indicating that autophagy signaling is crucial in modulating the inflammatory response. Normally, autophagy maintains cellular homeostasis by degrading damaged organelles and proteins, preventing inflammatory signal accumulation. When autophagy is inhibited, these signals accumulate, activating pathways that affect immune cell interactions. In cancer, this inflammatory response can enhance immune cell recruitment and activation, altering the tumor microenvironment and potentially affecting tumor progression and therapy response. Indeed, single-cell RNA-Seq demonstrated that ATG9A KO in cancer cells promoted macrophage activation towards a cytotoxic phenotype, characterized by markers such as *Inos*, *S100a8*, *Lcn2*, and *Cox-2*. These markers collectively highlight the role of oxidative stress as drivers of cytotoxic activity of macrophages.

The targetable macrophage marker CSF1R is highly expressed in regulatory macrophages. When we combined Trastuzumab with a CSF1R inhibitor in vivo in NSG mice (which lack NK cells and T-cells but not macrophages), no control tumors completely regressed, while the overwhelming majority (28/30) of ATG9A KO tumors completely regressed by the end of the treatment. Depleting the macrophage population with clodronate abrogated this effect. By enriching for cytotoxic, membrane-damaging macrophages, a robust and sustained anti-tumor response was obtained without the help of other immune cells, which may be particularly relevant to immunologically cold tumors. These findings highlight the crucial role of CSF1R in maintaining an immunosuppressive microenvironment and demonstrate that its inhibition, in conjunction with Trastuzumab, can significantly enhance therapeutic outcomes in ATG9A-deficient tumors.

TCGA analysis shows lower ATG9A mRNA is associated with better survival on the pan-cancer level, as well as in a number of individual cancers such as ovarian cancer. We focused on ovarian cancer cell lines, with in vivo studies using SKOV3 ovarian cancer cells, and supporting in vitro data from both SKOV3 and OVCAR-8 ovarian cancer cells (Supplementary Data 1). Our studies did not use syngeneic tumors, primary human PDX models, or non-ovarian cancer models, but future studies in these models would help elucidate the impact of other immune cells in ATG9A deficient tumors, cancer heterogeneity, and ATG9A's function in other tumor types. To enable the feasibility of the large-scale studies performed here, many of our in vitro studies were done using THP-1 cells with confirmation of key results using

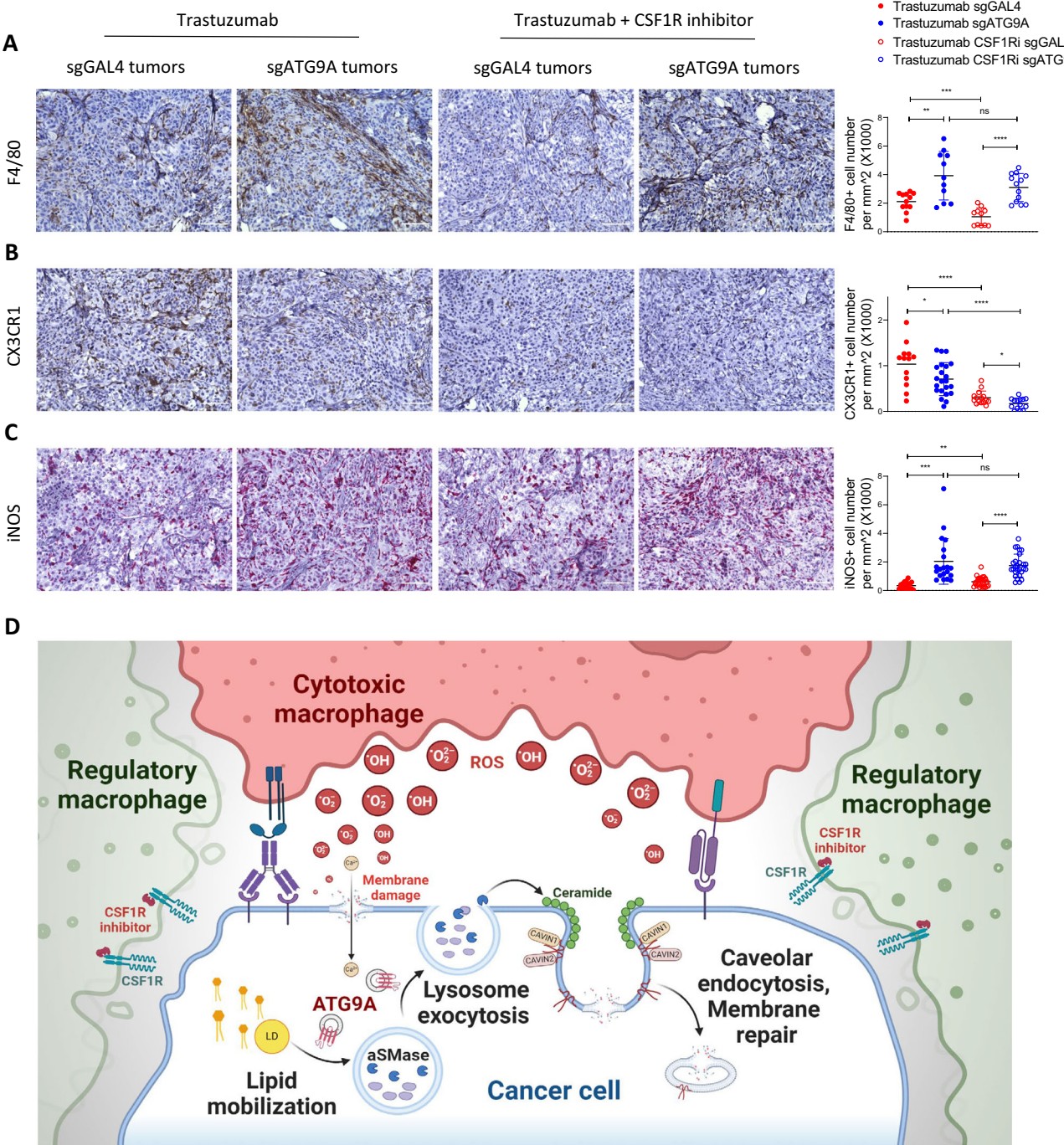

**Fig. 10 | CSF1R inhibition enhances cytotoxic macrophage recruitment in ATG9A-deficient ovarian tumors. A–C** Immunohistochemistry (IHC) staining for F4/80(pan-macrophage marker), CX3CR1 (regulatory macrophage marker), and iNOS (cytotoxic macrophage marker) in sgGAL4 (control) and sgATG9A (KO) tumors treated with either trastuzumab alone or in combination with a CSF1R inhibitor. (*n* = 15 biological replicates; mean ± SEM). Statistical significance was determined using two-way analysis of variance (ANOVA) followed by Tukey's post-hoc test. Scale bar = 100 μm. **D** Normally, macrophage-mediated cancer cytotoxicity is limited because tumor-derived cytokines polarize macrophages towards a pro-tumor regulatory phenotype. Cancer cells resist macrophage cytotoxicity through a plasma membrane repair mechanism characterized by ATG9A-dependent lysosome exocytosis, aSMase-mediated ceramide production, and caveolar endocytosis. ATG9A-mediated lipid mobilization from lipid droplets provides essential lipids for plasma membrane repair. ATG9A ablation in tumor cells skews their cytokine production and polarizes macrophages towards a ROS-producing, cytotoxic phenotype. These cytotoxic macrophages induce plasma membrane damage and lipid peroxidation in cancer cells. Without ATG9A-mediated repair, accumulation of plasma membrane damage ultimately results in cancer cell death. *$P < 0.05$; **$P < 0.01$; ***$P < 0.001$. Some elements of this figure were created with BioRender.com and are included under a publication license in accordance with BioRender's user agreement. *Created in BioRender. DeSelm, C. (2025)* https://BioRender.com/pkz1u1k.

primary human macrophages as well as xenograft mouse models for in vivo studies. However, a caveat of this approach is that THP-1 cells may not approximate primary macrophages in every in vitro condition tested.

As summarized in Fig. 7, our study reveals that ATG9A plays a multifaceted role in regulating ovarian cancer cell responses to macrophage-induced cytotoxicity through mechanisms involving plasma membrane repair, lipid metabolism, and inflammatory

signaling. Our findings show that ATG9A knockout sensitizes cancer cells to macrophage-mediated killing, specifically due to insufficient repair of ROS-induced plasma membrane damage, providing novel insights into the mechanism by which macrophages directly kill cancer cells. Additionally, the tumor regression observed with the combination of trastuzumab and CSF1R inhibition suggests that simultaneously targeting cancer cells and modulating the myeloid compartment may enhance macrophage-mediated cytotoxicity. These insights provide a strong rationale for further investigation of ATG9A-targeted approaches as a means to augment macrophage-based therapies, and to study its role in other tumor types, in the comprehensive fight against cancer.

# Method

## Cell lines and reagents

Most cell lines used in this study were originally purchased from the American Type Culture Collection (ATCC). Raji cell line was obtained from UCSF cell culture facility. THP-1, OVCAR-8, SKOV-3, T-24 and Raji cells were grown in RPMI 1640 medium (Gibco, A1049101) supplemented with 10% fetal bovine serum (FBS; Gibco, 26140079) and 1X Pen-Strep (Gibco, 15140-122). 293 T cells were grown in DMEM medium with 10% FBS and 1X Pen-Strep. Cells were grown in a 5% CO2 humidified incubator at 37 °C. Cell line STR authentications were performed at the UC Berkeley DNA Sequencing facility. All cell lines used in this study were tested for Mycoplasma using the Lonza MycoAlert Detection Kit (Lonza, LT07-318) and control set (Lonza, LT07-518).

## THP-1 macrophage differentiation

THP-1 cells were cultured in RPMI medium supplemented with 50 ng/ml of phorbol 12-myristate 13-acetate (PMA; Sigma-Aldrich, P8139) for a duration of 72 h. Following PMA treatment, the media was replaced with fresh RPMI containing 25 ng/ml of interleukin-4 (IL-4; R&D Systems, 204-IL-010) for an additional 48 h.

## CAR-THP-1 generation

Both the α-EphA2 CAR and α-CD19 CAR plasmids were generously provided by Dr. Carl DeSelm at Washington University in St. Louis. To generate retrovirus, 1.4 μg of CAR plasmid and 0.6 μg of the pCL-Ampho packaging plasmid were added to Opti-MEM (Gibco, 31985062) along with 15 μl of Mirus LT1 reagent (Mirusbio, MIR2304), and the mixture was incubated for 30 min at room temperature. 500,000 293 T cells were seeded in each well of a six-well plate and infected for 2 days. The viral media was then collected, filtered, and transduced into THP-1 cells in the presence of 8 μg/ml polybrene using a spin infection protocol (800 g, 1 h 30 min, 32 °C). Confirmation of CAR expression was achieved by flow cytometry staining using an FMC63 scFv recognizing antibody (Acrobiosystems, FM3-HPY53) for α-CD19 CAR-THP-1 and F(ab')₂ fragment recognizing antibody (Jacksonimmuno, 115-606-072) for α-EphA2 CAR-THP-1 respectively.

## Human CAR-Ms production

Peripheral blood mononuclear cells (PBMCs) were obtained from STEMCELL and subjected to EasySep Human CD14 positive selection kit II according to the manufacturer's protocol. To differentiate PBMCs into Human monocyte-derived macrophages, two million isolated CD14 monocytes were cultured in a 6-well plate in RPMI with 10% FBS, 1% penicillin-streptomycin, 1X GlutaMAX, 1X HEPES and 20 ng/ml recombinant human GM-CSF (PeproTech) for 9 days. The Ad5f35 virus was introduced on day 5 at a multiplicity of infection (MOI) of 0.5 ×103 based on PFU titer. On day 9, the differentiated macrophages were harvested and analyzed for GFP and CAR expression using flow cytometry. PBMCs or CD14+ monocytes were cryopreserved in RPMI culture medium containing 20% FBS, 10% DMSO. Cryopreserved cells were carefully thawed and washed with EasySep buffer (PBS + 2% FBS) prior to further use. The Ad5f35 vector utilized in our study to

overexpress HER2-ζ, pAd5/F35[Exp]-EGFP-EF1A>Her2CD3Z, was constructed and packaged by VectorBuilder (vector ID: VB230822-1453hbw). Detailed information on this vector can be found on vectorbuilder.com.

## CRISPR screen

The OVCAR-8 cell line exhibits high levels of EphA2 expression. To induce CD19 overexpression, lentiviral vectors carrying CD19 overexpression plasmids (kindly provided by Dr. Carl DeSelm's lab) were used to infect OVCAR-8 cells. High CAR antigen expression was confirmed on OVCAR-8 CD19⁺ cells before the screen. For the CRISPR screen, OVCAR-8 cells or CD19⁺ OVCAR-8 cells, were infected with the sgRNA library at a multiplicity of infection <0.3. Post-infection, cells underwent puromycin selection (5 μg/ml) for 3 days, followed by a one-day recovery period in normal growth medium without puromycin. Sufficient sgRNA library representation was confirmed by deep sequencing after selection. CAR-THP-1 macrophages were directly added to the culture at a 2:1 effector-to-target ratio. Throughout the screen, 1000X coverage was maintained. Samples were collected at T0 and at day-6. Genomic DNA was isolated from these samples and subjected to next-generation sequencing. Following data acquisition, the FASTQ files from the CRISPR screen were trimmed for adapters using cutadapt, and then aligned to the sgRNA library using MAGeCK count with default parameters. Lastly, CRISPR screen hits were identified and confirmed using the MAGeCK FluteMLE pipeline.

## Pooled sgRNA library design

The immune surface sgRNA sublibrary is comprised of 8,240 sgRNAs targeting surface and immune related genes. This library consists of 5 sgRNAs targeting each of 1378 experimental genes, 70 essential genes, 70 non-essential genes and 60 immune-related genes, plus 350 non-targeting sgRNA controls.

The experimental genes were selected from a curated list of experimentally validated or in silico predicted surface proteins[42–45] and further filtered based on a surface localization confidence score as well as a combined protein and RNA expression score derived from in-house data. The essential control genes were selected from Depmap common essential genes, the Achilles CRISPR dataset, and our in-house common essential gene list. Olfactory receptor genes were selected as non-essential control genes. Immune-related control genes were curated based on the literature, which included genes encoding for both surface and intracellular proteins. Specific sgRNA sequences were designed using the sgRNA designer tool from Broad Institute. The 350 non-targeting control sgRNAs were selected from library A of the human GeCKOv2 library[46].

## Pooled sgRNA library cloning

The immune surface sgRNA sublibrary is cloned into the lentiCRISPRv2 backbone following a modified version of LentiCRISPRv2 cloning protocol from MIT. Briefly, the lentiCRISPRv2 (Addgene #52961) was linearized through Excel restriction enzyme digestion followed by gel extraction cleanup. The sgRNA library oligo was synthesized through Twist Biosciences, diluted to 10 nM and PCR amplified using HF Phusion (NEB #M0530L). Amplified library oligos were purified using QIAgen MiniElute purification kit (QIAgen #28004) and subsequently cloned into the linearized lentiCRISPRv2 vector through golden gate reaction at 3:1 sgRNA oligo:vector ratio. The ligated pooled library was purified through traditional DNA precipitation and extraction methods prior to transformation into high efficiency chemically potent Stellar cells (Takara #636736) for transformation efficiency testing and MegaX DH10B electrocompetent cells (ThermoFisher #C640003) for scale-up sample preparation. Transformed bacteria cells were grown in liquid LB broth containing selection drug at 37 °C for <16 h. Amplified sgRNA library pool was extracted using Maxi plasmid extraction kits

(QIAgen #12162), and subsequently sent for NGS on the Illumina MiSeq platform for verification.

## Individual sgRNA KO

Individual sgRNA KO was achieved using the lentiCRISPRv2 plasmid. Lentiviral vector digestion, oligo annealing, and cloning into the digested vector were performed following the Zhang Lab protocol. After confirming the plasmid sequence through Sanger sequencing, the plasmids were packaged into lentivirus using a previously established protocol. Briefly, three million 293 T cells were seeded into 10 cm plates the day before infection. On the day of infection, 3.2 µg of CRISPR KO plasmids, 3.2 µg of dR8.91 plasmids, and 0.4 µg of pMD2.G were added to Opti-MEM (Gibco, 31985062) along with 19.2 µl of Mirus LT1 reagent (Mirusbio, MIR2304). The mixture was then incubated for 30 min at room temperature before being added dropwise to the 293 T cells. The cells were cultured for 3 days, after which the media was collected and filtered.

## Plasmids, siRNAs and transfection

ATG9A-RFP plasmid was a gift from Noboru Mizushima (Addgene #60609), CAVIN2-EGFP plasmid was a gift from Ari Helenius (Addgene #27710), MyrPalm-EGFP plasmid was a gift from Daniel Gerlich (Addgene #21037). Silencer siRNAs targeting SMPD1 (siRNA ID: s13169) and SMPD2 (siRNA ID: 12495) and scramble Silencer siRNA control was purchased from Thermo Fisher Scientific (Rochester, NY). Briefly, OVCAR-8 cells were first seeded in 6 cm culture dishes to reach 60–70% confluence, then 25 pmol of siRNA and 7.5 µl of Lipofectamine® RNAiMAX (Thermo Fisher Scientific, 13778075) in 500 µl of Opti-MEM (Gibco, 31985062) were added for a 24-h incubation.

## IncuCyte co-culture assay

Cancer cells were labeled using NucLight Red lentivirus (Sartorius, 4476) for optimal results in the IncuCyte experiment. One day before imaging, cancer cells were seeded at density of 5,000 cells in 50 µl per well in Falcon 96 well plate (Corning, 353072). On the imaging day, 50 µl of media containing the CAR-THP-1 macrophages were added to each well at 2:1 E:T ratio. The result at day three were normalized by the count of the first scan. The live cell percentage was calculated by dividing the result from co-culture wells by cancer cell only control. Statistical analysis was performed using GraphPad Prism. Drugs/inhibitors/neutralizing antibodies used in the IncuCyte co-culture assay are listed in the Supplementary data.

## IncuCyte 3D co-culture assay

10,000 NucLight Red+ OVCAR-8 cells were either cultured alone or mixed with 20,000 GFP-labeled α-EphA2-CAR-THP-1 macrophages and seeded into each well of 96-well Round Bottom Ultra-Low Attachment Microplate (Corning, 7007). The red fluorescence intensity of each well at day three was normalized by the red fluorescence intensity of the first scan. Red mean fluorescence (MFI) intensity in the co-cultured group was normalized to the cancer-cell-only group. Data was quantified using Excel. Statistical analysis was performed using GraphPad Prism 9.2.0.

## PM integrity assay

In the co-culture experiment, cancer cells and GFP-labeled α-EphA2-CAR-THP-1 macrophages were mixed at a 1:1 E:T ratio in RPMI media and incubated in a 1.5 Eppendorf tube at 37 °C for 2–4 h. A brief spin (200 g, 1 min) was conducted to facilitate cell-cell contact. Subsequently, the cells were centrifuged again at 300 g for 5 min, and the resulting pellet was resuspended in FACS buffer (2% FBS in PBS) containing 1 µg/ml PI for flow cytometry analysis. After a 5-min incubation, the samples were analyzed using the Attune flow cytometer. Cancer

cells were identified based on their negative GFP expression. Data analysis was performed using FlowJo 10.7.2, and statistical analysis was conducted using GraphPad Prism 9.2.0.

## Phagocytosis assay

In the co-culture experiment, NucLight Red-labeled cancer cells and GFP-labeled EphA2-CAR-THP-1 macrophages were mixed at a 1:1 E:T ratio in RPMI media and incubated in a 1.5 Eppendorf tube at 37 °C for 2–4 h. A brief spin (200 g, 1 min) was conducted to enhance cell-cell contact. Subsequently, the cells were centrifuged again at 300 g for 5 min and resuspended in FACS buffer (2% FBS in PBS) for analysis using the Attune flow cytometer. Data analysis was performed using FlowJo 10.7.2, where GFP+ cells were first gated, followed by calculation of the proportion of NucLight Red+ cells among GFP+ cells to determine the phagocytosis rate. Statistical analysis was conducted using GraphPad Prism 9.2.0.

## HaloTag plasma membrane integrity assay

OVCAR-8 cells were transfected with the HT probe (PEX3–GFP–HT, Addgene #67764) and co-cultured with BFP-labeled macrophages. Cells were incubated with complete medium containing the membrane impermeant ligand Alexa Fluor 660 (MIL) and membrane permeant ligand TMR (MPL) (1:1000) for 1 min before 4% PFA fixation.

For quantification, cells transfected with the HT probe were gated using GFP total cell fluorescence intensity. The fluorescence intensity of MIL or MPL colocalizing with GFP–HT masks (GFP$^+$MIL$^+$ or GFP$^+$MPL$^+$, respectively) was then assessed.

## PHrodo phagocytosis assay

Cancer cells were labeled with pHrodo Red (Invitrogen, P36600) according to the manufacturer's instructions. Following thorough washing, the pHrodo Red-labeled cancer cells were co-cultured with EphA2-CAR-macrophages at a 1:1 ratio. Red fluorescence was then measured and quantified using the IncuCyte live-cell imaging system.

## In vitro ADCP assay

Cancer cells were pre-opsonized by incubating with 10 µg/mL Trastuzumab. SKOV-3 cells, a HER2+ cell line, were used as target cells; OVCAR-8 cells, a HER2- cell line, were used as negative control cell line. THP-1 macrophages were co-cultured with cancer cells at a 2:1 ratio. The number of cancer cells was recorded and quantified over time using the IncuCyte live-cell imaging system.

## Co-culture with or without Ca$^{2+}$

Co-culture of OVCAR-8 cells and α-EphA2-CAR-THP-1 macrophages were performed either using regular DMEM (Gibco, 11960044) or Ca$^{2+}$ free DMEM (Gibco, 21068028) for 2 h. PM integrity assay was performed as described above.

## Competition assay

We first infected OVCAR-8 and SKOV-3 cells with NucLight Red or Green lentivirus (Sartorius, 4476, 4475). And sorted cells with very high fluorescence intensity. Subsequently, NucLight Green cells were infected with control guides (sgGAL4), while NucLight Red cells were infected with test guides. Three days of puromycin selection was performed. A mixture of 100,000 cancer cells with control guides and 100,000 cancer cells with test guides was seeded into one well of a six-well plate. Additionally, 400,000 THP-1 macrophages were added to each well and cultured for 3 days. Cells were then trypsinized and collected for flow cytometry analysis to quantify the ratio of NucLight Red and Green positive cells. Subsequently, 1/4 of the cells were seeded back into the well with another 400,000 THP-1 macrophages added and cultured for an additional 3 days. At the end of the co-culture, the cells were trypsinized and collected for flow cytometry analysis.

## Nanolive co-culture experiment

On the day before imaging, 50,000 OVCAR-8 cells were seeded onto a 35 mm μ-Dish (ibidi, 81156). Two hours before imaging, 50,000 EphA2-CAR-THP-1 macrophages were added into the dish. Subsequently, the dish was imaged and quantified utilizing a Nanolive live cell imaging platform (Nanolive, 3D Cell Explorer 96focus). Quantification was performed using Nanolive software, statistical analysis was performed using GraphPad Prism 9.2.0.

## Indirect co-culture

Indirect co-culture was conducted by seeding 50,000 EphA2-CAR-THP-1 macrophages on top of a 0.4 μm membrane insert (Corning, 3450), while 50,000 OVCAR-8 cells were cultured at the bottom of a 12-well plate. For direct co-culture control, 50,000 CAR-THP-1 macrophages and OVCAR-8 cells were mixed and cultured in a regular 12-well plate. Following a 3 day co-culture period, cell counting was carried out, and the live cell percentage was determined by dividing the cell number in the co-cultured wells by the cell number in the cancer-only wells.

## Conditioned media generation

Conditioned media was generated by seeding three million OVCAR-8 cells in a 10 cm dish containing 10% FBS RPMI media. The following day, the media was centrifuged to remove cell debris. Conditioned media was collected, aliquoted and stored in −80 °C.

## Cytokine array

The conditioned media was generated as described above. The Human Cytokine Array C5 (Raybiotech, AAH-CYT-5-8) was conducted following the manufacturer's instructions. Conditioned media from OVCAR-8 sgGAL4 and sgATG9A cells were utilized to incubate the cytokine array membranes overnight. Subsequently, the membranes were exposed to Biotinylated Detection Antibody Cocktail followed by HRP-Conjugated Streptavidin and then imaged using a chemiluminescent imaging system. Two biological replicates were performed for each sample, and all dot intensities were normalized using positive and negative controls.

## Flow cytometry

For flow cytometry staining, cells were harvested and suspended in FACS buffer containing the appropriate primary antibody, then incubated on ice for 30 min. Details of the flow cytometry antibodies used are provided in Supplementary data. If the primary antibody was not fluorescent-conjugated primary antibodies, the samples were washed and suspended in FACS buffer containing Alexa Fluor 488, 594, or 647-conjugated anti-rabbit or anti-mouse secondary antibody. The samples were then incubated on ice for an additional 30 min. Following incubation, the samples were washed twice to remove any unbound antibodies. Finally, the stained samples were loaded onto the Attune flow cytometer for analysis. All wash steps were performed in FACS buffer. Data analysis was performed using FlowJo 10.7.2, and statistical analysis was conducted using GraphPad Prism 9.2.0.

## BODIPY lipid droplet assay

Cells were seeded in Nunc Lab-Tek II Chamber Slide (Thermo Scientific 154534) the day before staining. BODIPY 493/503 (Invitrogen, D3922) was diluted in serum-free RPMI medium to a concentration of 10 μM, and the cells were incubated with the staining solution for 15 min at 37 °C. After staining incubation, cells were fixed using 4% Paraformaldehyde (Thermo Fisher Scientific, J61899). Imaging was performed using a Zeiss Spinning Disk Confocal microscope. Lipid droplet quantification was carried out using QuPath 0.4.4, and data analysis was performed using GraphPad Prism 9.2.0.

## RNA-extraction and qPCR

RNA extraction from the cells followed the manufacturer's protocol using the Quick-RNA extraction kit (Zymo, R1054). Subsequently, cDNA was synthesized using the SuperScript III First-Strand Synthesis System (Invitrogen, 18080). The mRNA expression levels of the genes were quantified using primers listed in the Supplementary data and SYBR Green Real-Time PCR Master Mixes (Thermo Fisher Scientific, A25742) on the QuantStudio Flex Real-Time PCR system.

## Western blot and immunoprecipitation

Toatl Protein was extracted from cells using RIPA buffer (Thermo Fisher Scientific, 89900) and protease inhibitor (Thermo Fisher Scientific, 78430) 1x cocktail. Antibody information is listed in the Supplementary data. Membranes were incubated with primary antibodies overnight at 4 °C, washed three times using TBST (Tris-buffered saline, 0.1% Tween 20), and then incubated with IRDye 800CW donkey anti-rabbit IgG secondary antibody (LI-COR Biosciences, 926-32213, 1:2000) or IRDye 680RD donkey anti-mouse IgG secondary antibody (LI-COR Biosciences, 926-68072, 1:2000). The membranes were then washed again using TBST and processed for scanning and visualization using an Odyssey CLx imaging system (LI-COR, model #9140). Protein bands were quantified using ImageJ.

## Co-immunoprecipitation (co-IP)

Co-IP was performed using a Pierce Crosslink Immunoprecipitation Kit according to the manufacturer's instruction (Thermo Fisher Scientific, 26147). Briefly, 10 μg of ATG9A antibody (Cell Signaling Technology, 13509S) or rabbit IgG (Thermo Fisher Scientific, 31903) was added to a spin column containing 20 μl of resin slurry in a 2 ml collection tube. The mix was incubated at room temperature with gentle rotation for 1 h. After incubation, the spin columns were centrifuged, and the antibody-bound resin was rinsed with 1X coupling buffer three times to remove unbound antibodies. 50 μl of 2.5 mM disuccinimidyl suberate (DSS) crosslinker solution was added to crosslink the antibody to the resin. Subsequently, 750 μg of protein extract was added to the spin column and rotated end-over-end at 4°C overnight. Post-incubation, the column was washed three times with 1X TBS and one time with 1X conditioning buffer. 50 μl of elution buffer was used to elute the samples from the column. The eluate was boiled with sample buffer at 95°C for 5 min for Western blotting.

## Immunofluorescence staining

One day prior to staining, cells were seeded into Nunc Lab-Tek II Chamber Slide (Thermo Scientific 154534). On staining day, cells were fixed with 4% paraformaldehyde (Thermo Fisher Scientific, J61899) for 15 min at room temperature then permeabilized using 0.1% Triton X-100 (Thermo Fisher Scientific, 85111) for 10 min on ice. After three washes with PBS, cells were blocked with 10% bovine serum albumin (BSA) for 1 h. Primary antibodies, as detailed in the Supplemental data, were added to specific chambers and incubated overnight at 4°C. The next day, cells were washed three times with PBS and incubated with Alexa Fluor 488, 594, or 647-conjugated secondary antibodies (Invitrogen, Carlsbad, CA) to visualize protein expression. Slides were then mounted using VEC-TASHIELD Antifade Mounting Medium (Vector Laboratories, H-1900), cover slipped and imaged using a Zeiss Spinning Disk Confocal microscope. Quantification of images was performed using ImageJ or ZEN 3.9. Data analysis was performed using GraphPad Prism 9.2.0.

## Confocal imaging

Confocal imaging of live cells labeled with various plasmids was conducted using a Zeiss Spinning Disk Confocal microscope. Unless otherwise specified, images were captured at 64X magnification. Image analysis was performed using ZEN 3.9 software. Mean fluorescence quantification was conducted by contouring on cell border and quantifying the fluorescence intensity. For colocalization analysis, individual cells were outlined, and Pearson's correlation coefficient was generated and recorded. Data analysis was carried out using GraphPad Prism 9.2.0.

## Animal monitoring and endpoints

All animal experiments were conducted in compliance with institutional guidelines approved by the Institutional Animal Care and Use Committee (IACUC)] at UCSF. All animals were housed in a specific pathogen-free (SPF) facility at the University of California, San Francisco (UCSF). Experimental and control groups were housed in the same room but in separate cages and were not co-housed, to avoid cross-exposure to experimental conditions. The facility maintained controlled temperature and humidity with a 12 h light/dark cycle.

The maximum allowable tumor size was 2 cm in any dimension, in accordance with institutional ethical standards. Tumor size was measured 2 weekly using calipers, and tumor volume was calculated using the formula: (width × length$^2$) × 0.52. For metastasis tumor model, bioluminescence imaging was performed weekly to monitor tumor progression. Animals were monitored at least twice weekly for signs of distress, including respiratory difficulty, hunched posture, body condition score of 2 or lower, impaired or decreased mobility, neurological signs, reduced grooming, and the presence of lesions unresponsive to treatment. Animals meeting humane endpoints—including tumor size exceeding 2 cm, tumor ulceration, interference with normal functions, or significant clinical deterioration—were euthanized using carbon dioxide ($CO_2$) inhalation followed by confirmation of death.

In immunocompromised mice, weight loss was not used as a sole criterion for euthanasia due to their susceptibility to transient diarrhea and associated weight fluctuations. If weight loss was noted, veterinary staff were consulted to initiate supportive interventions including provision of diet gels, supplemental feeding, and intraperitoneal saline injections. If weight loss persisted and reached ≥ 20% of the initial body weight at the time diarrhea was first observed, animals were humanely euthanized after consultation with veterinary staff.

## Subcutaneous SKOV3 tumor model

Animal experiments were conducted following protocols approved by the Institutional Animal Care and Use Committee (IACUC) at UCSF. Female NSG (NOD.Cg-Prkdc^scid Il2rg^tm1Wjl/SzJ) mice (Jackson Laboratory, Strain #:005557), aged 6–8 weeks, were used for tumor implantation.

For the subcutaneous tumor model, two million sgGAL4 or sgATG9A SKOV3 cells were subcutaneously injected into the flanks of NSG mice. Once tumors reached ~50 mm$^3$, mice were randomized into treatment groups.

Mice received intraperitoneal (IP) injections of Trastuzumab (Bio X cell, SIM0005) or human IgG isotype control (Bio X cell, BE0297) at a dose of 10 mg/kg every 2–3 weeks. Some mice also received 100 μl Clodrosome (Encapsula NanoSciences, CLD-8909) twice a week to deplete macrophages in tumors. For CSF1R inhibitor treated mice, mice were fed with a diet containing the CSF1R inhibitor PLX5622 (MedChemExpress, HY-114153) at a concentration of 1200 ppm, which corresponds to an estimated mean plasma concentration of 9090 ng/mL. The diet was provided by Research Diets, USA.

After 6–7 weeks of treatment, mice were euthanized, and tumors were harvested for downstream analyses, including histological examination, immunohistochemistry, and other molecular assays. Data was analyzed using GraphPad Prism software.

## Intraperitoneal SKOV-3 tumor model

We injected 100,000 sgGAL4 or sgATG9A SKOV3-luciferase cells intraperitoneally (IP) into NSG mice. Five days post-tumor injection, tumors were established, and bioluminescence imaging was initiated using the IVIS system twice a week. Mice received a weekly IP injection of five million EphA2-CAR-M cells. Total flux of luciferase signals was measured and plotted to assess tumor burden and CAR-M cytotoxicity.

## Immunohistochemistry

Immunohistochemistry staining was conducted following established protocols. Slides were preheated until paraffin was melted (52°C, 30 min), followed by deparaffinization and rehydration. Antigen retrieval was performed using either Citrate or Tris EDTA. After PBS washing, slides were incubated overnight at 4°C in a humidified chamber with primary antibodies. The following day, slides were washed with TNT buffer (0.1 M Tris.HCl pH 7.5, 0.15 M NaCl, 0.05% Tween-20) and secondary HRP polymer conjugated Fab' anti-mouse IgG(H + L) or anti-Rabbit AP Polymer were applied and incubated for 1 h. Following another wash with TNT buffer, DAB or AP working solution was added to cover the tissue completely and incubated until desired staining intensity was achieved. Nuclear staining was performed with hematoxylin. Sections were dehydrated through a series of alcohol washes and cleared with xylene. Finally, sections were mounted with mounting medium and allowed to dry completely. Imaging was performed using a Keyence microscope (BZ-X800). Quantification was performed using QuPath 0.4.4 and data analysis was performed using GraphPad Prism 9.2.0.

## RNA-Seq

The RNA-seq paired end FASTQ data generated by Illumina HiSeq 4000 sequencing system were first trimmed to remove adapter sequences using Cutadapt v2.6 with the "-q 10 -m 20" option. After adapter trimming, FASTQC v0.11.8 was used to evaluate the sequence trimming as well as overall sequence quality. Using the splice-aware aligner STAR (2.7.1a), RNA-seq reads were aligned onto the Human reference genome build GRCh38decoy using the "--outSAMtype BAM SortedByCoordinate --outSAMunmapped Within -- outSAMmapqUnique 50 --sjdbOverhang 65 --chimSegmentMin 12 --twopassMode Basic" option and exon-exon junctions, with Human gene model annotation from GENCODE v28. Gene expression quantification of uniquely mapped reads was performed using "featurecount" function within Rsubread R-package with "GTF.featureType = "exon", GTF.attrType = "gene_id", useMetaFeatures=TRUE, allowMultiOverlap=FALSE, countMultiMappingReads=FALSE, isLongRead=FALSE, ignoreDup=FALSE, strandSpecific=0, juncCounts=TRUE, genome=NULL, isPairedEnd=FALSE, requireBothEndsMapped=FALSE, checkFragLength=FALSE, countChimericFragments=TRUE, autosort=TRUE" option. Cross-sample normalization of expression values and differential expression analysis was done using DESeq2 R-package. Benjamini-Hochberg corrected $p$-value < 0.05 and log2 foldchange > 1 or < −1 were considered statistically significant. For RNA-seq, gene set enrichment analyses were performed using the pre-ranked method implemented in the fgsea R package (10.18129/B9.bioc.GSEABase), and Hallmark gene sets were downloaded from the molecular signatures database (MSigDB), genes were ranked by the Wald-statistics from DESeq2.

## Single cell library preparation and RNA sequencing

Tumors were collected after 6–7 weeks of treatment, weighed, and processed to obtain single-cell suspensions. For every gram of tumor tissue, 5 mL of tumor digestion medium was prepared by combining 500 μL Collagenase/Hyaluronidase, 750 μL DNase I solution (1 mg/mL), and 3.75 mL RPMI 1640 medium, mixed thoroughly, and warmed to room temperature. Harvested tumor tissue was minced into small pieces (≤ 2 mm), transferred to tubes containing digestion medium, and incubated at 37°C for 20–30 min on a shaking platform (500 rpm) with intermittent pipetting. The digested tissue was filtered through 70 μm and 40 μm nylon mesh strainers, pelleted at 300 x g for 10 min, and resuspended in 1–2 mL PBS. Cells were stained for 1 h with LIVE/DEAD Fixable Green Dead Cell Stain (L23101, Invitrogen) and a mouse CD45 antibody (Biolegend, 157212) before being run on a Fusion 1 cell sorter to isolate the CD45 + /dead cell stain- cells. Once sorted, samples were ready for single-cell Isolation and subsequent RNA sequencing.

Single cells were isolated on the Chromium X using a Chip G with a targeted cell recovery of 10,000 per sample. Single cell RNA-Seq libraries were prepared using the 10x Genomics Next GEM Single Cell 3' v3.1 chemistry following manufacturer's protocols. Libraries were quantified using the High Sensitivity D1000 reagents on a TapeStation 4200 and sequenced on the Illumina NovaSeq X to a targeted depth of ~20,000 reads per cell.

### Single-cell RNA sequencing analysis and visualization

The demultiplexing, barcode processing, gene counting, and aggregation were made using the Cell Ranger software v8.0.0. Briefly, data demultiplexing and alignment were performed using Cell Ranger. The resulting FASTQ files were aligned to the mouse reference genome (mm39). The cellranger aggr pipeline was used to aggregate the gene-barcode matrices. This allowed for normalization and integration of data from all samples.

The barcode-gene matrices from the Cell Ranger pipeline were further analyzed using the Loupe Browser v8.0.0. Non-macrophage cells were filtered out by Log2 CD14 expression <2. Following standard practices to exclude low-quality cells (UMIs per barcode (Log2) between 10.5-15.5; Genes per barcode (Log2) between 9.5-12.5; Mitochondrial UMIs per barcode below 15%), 6413 cells from sgGAL4 tumors and 6996 cells from sgATG9A tumors were identified from the datasets. Cluster identification and all differential expression analyses were performed using Loupe Browser v8.0.0. We selected $k = 2$ for k-means clustering, as this configuration yielded the smallest inter-cluster distance, indicating optimal separation between the clusters. The expression levels of specific genes of interest were visualized across different cell populations using violin plots, feature plots, and heatmaps generated within Loupe Browser.

### Lipidomics

For our lipidomics sample preparation, NucLight Red-labeled sgGAL4 or sgATG9A cancer cells were cultured alone or co-cultured with α-EphA2-CAR-THP-1 macrophages for 3 days. Co-cultured samples were sorted twice based on NucLight Red expression, with ~30−50 million cells used for sorting and 15 million cells sorted based on NucLight Red expression in cancer cells. Plasma membrane samples were isolated using a cell fractionation kit from Invent Biotechnologies (SM-005).

Untargeted lipidomics was performed by MTAC@MGI at Washington University in Saint Louis, with two replicates for each sample. Lipids were extracted using MTBE (methyl tert-butyl ether) via tip sonication and vortex mixing. The upper phase containing lipids was collected and dried by SpeedVac without heat. Dried samples were reconstituted with 30 μL of 50% EtOH and subjected to LC-MS/MS. LC conditions utilized the Vanquish Horizon UHPLC System (Thermo Scientific). MS2 spectra was obtained with Orbitrap Tribrid ID-X mass spectrometer (Thermo Scientific) accompanied by AcquireX Deep Scan technology. Database search was conducted using Compound Discoverer 3.3 SP2. Additional statistical analysis was performed using R script software.

### Preparation of shotgun proteomics samples

OVCAR-8 cells were harvested from plates using collagenolytic solution (Accutase, Stemcell, 07920) to preserve cell surface epitopes. Following three washes with PBS, cells were aliquoted to 3 million cells per replicate and subjected to reduction and alkylation in lyse buffer (PreOmics iST Kit). Subsequently, DNA was sheared via sonication, and the samples were centrifuged at 21,000 x g for 10 min to clear the lysate. Upon the addition of trypsin (PreOmics), samples were incubated at 37 °C for 1.5 h before undergoing cleanup with C18 columns following the PreOmics kit protocol. Following elution, the samples were vacuum dried using centrifugation (CentriVap, Labconco) and dissolved in PreOmics LC-load solution (2% acetonitrile, 0.1% TFA). Peptides were quantified using the Pierce Quantitative Colorimetric Peptide Assay (Thermo Fisher Scientific, 23275) and resuspended to 200 ng/μL for mass spectrometry analysis.

### Preparation of cell surface proteomics samples

OVCAR-8 cells were harvested from plates with collagenolytic solution (Accutase, Stemcell, 07920) to preserve cell surface epitopes. After washing 3x with PBS, cells were aliquoted to three million cells per replicate in 500 μL PBS. Cell surface glycans were oxidized by addition of sodium metaperiodate (1.6 mM), then incubated for 20 min at 4 °C on an end-to-end rotor. The oxidized cell surface moieties were then biotinylated (biocytin hydrazide, 1 mM) in PBS containing aniline (1 mM). Cells were washed 3X in PBS and snap-frozen in liquid nitrogen. After thawing, the cells were resuspended in 1X RIPA buffer and sonicated before a 2 h incubation with streptavidin beads (Pierce, High Capacity Neutravidin Beads) at 4 °C on an end-to-end rotor. The beads were washed sequentially with 5 mL buffer A (1X RIPA, 1 mM EDTA), buffer B (1X PBS, 1 M NaCl), and buffer C (2 M urea, 50 mM ABC). The washed beads containing biotinylated cell surface proteins were then resuspended in 4 M urea digestion buffer (50 mM tris pH 8.5, 10 mM TCEP, 20 mM iodoacetamide, 4 M Urea) and incubated for 10 min at 55 °C. After the addition of trypsin (PreOmics), the samples were left to incubate for 1.5 h at 37 °C to release peptides. The trypsinization was then halted by acidification with TFA (1%), and peptides were desalted with C18 columns according to the PreOmics kit protocol. After elution, the samples were vacuum dried with centrifugation and dissolved in PreOmics LC-load solution (2% acetonitrile, 0.1% TFA). Peptides were quantified using Pierce Quantitative Colorimetric Peptide Assay (Thermo Fisher Scientific, 23275) and resuspended to 200 ng/μL for mass spectrometry.

### LC-MS/MS

Liquid chromatography and mass spectrometry was performed as previously described[47]. Briefly, approximately 1 μL of peptides were separated using a nanoElute UHPLC system (Bruker) with a pre-packed 25 cm × 75 μm Aurora Series UHPLC column+ CaptiveSpray insert (CSI) column (120 Å pore size, IonOpticks, AUR2-25075C18A-CSI) and analyzed on a timsTOF Pro 2 (Bruker) mass spectrometer. Peptides were separated using a linear gradient of 2%−34% solvent B (solvent A: 2% acetonitrile and 0.1% formic acid; solvent B: acetonitrile and 0.1% formic acid) over 100 min at 400 nl/min. Data-dependent acquisition was performed with parallel accumulation-serial fragmentation (PASEF) and trapped ion mobility spectrometry (TIMS) enabled with 10 PASEF scans per topN acquisition cycle. The TIMS analyzer was operated at a fixed duty cycle close to 100% using equal accumulation and ramp times of 100 ms each. Singly charged precursors were excluded by their position in the m/z−ion mobility plane, and precursors that reached a target value of 20,000 arbitrary units were dynamically excluded for 0.4 min. The quadrupole isolation width was set to 2 m/z for m/z < 700 and to 3 m/z for m/z > 700 and a mass scan range of 100−1700 m/z. TIMS elution voltages were calibrated linearly to obtain the reduced ion mobility coefficients (1/K0) using three Agilent ESI-L Tuning Mix ions (m/z 622, 922, and 1222).

### LC-MS/MS data processing and analysis

The LC-MS/MS raw data (.d files) were processed using MSFragger within FragPipe with default Label Free Quantification (LFQ) settings and searched against the human UniProt database (accessed 04/15/23). The contaminant and decoy protein sequences were added to the search database via Fragpipe. Enzyme specificity was set to trypsin with up to two missed cleavages. Cysteine carbamidomethylation was set as the only fixed modification, while acetylation (N-term) and methionine oxidation were set as variable modifications. The

computational tools PeptideProphet and ProteinProphet were used for statistical validation of results and subsequent mapping of the peptides to the proteins respectively with 1% FDR. The LFQ intensities were considered for secondary data analysis. Missing values were imputed by KNN for samples with one missing value per group, while the binary proteins were left unimputed with the Differential Expression Proteomics (DEP) R package. Peak values were then log2(x) transformed, normalized and subjected to Welch's $t$-tests. Proteins with $p$-values <0.05 were used to identify the differentially expressed proteins. Statistical analysis and data visualization were performed in R studio, MetaboAnalyst 5.0[48], and Microsoft Excel.

### Statistical analysis and reproducibility

All statistical analyses were performed using GraphPad Prism software. Data are presented as mean ± standard deviation (SD) or mean ± standard error of the mean (SEM) as indicated. Statistical significance was determined using two-tailed unpaired Student's $t$-tests, one-way analysis of variance (ANOVA) followed by Tukey's post-hoc test, or two-way ANOVA followed by Sidak's multiple comparisons test as appropriate. The ovarian serous cystadenocarcinoma (TCGA PanCancer Atlas[49]) dataset was downloaded via cBioPortal[50]. RSEM (RNA-Seq by Expectation Maximization)[51] was used by the TCGA to quantify mRNA expression for this dataset. Survival analysis was performed using the survival package in R and survival data was visualized using the Kaplan-Meier method, with the endpoint being overall survival defined as the time interval from the time of tumor tissue acquisition to death from any cause. All statistical tests were two-sided unless otherwise specified, with a $p < 0.05$ considered statistically significant. All experiments were independently repeated at least three times with similar results. Representative data are shown, and where applicable, the number of biological replicates is provided in the figure legends.

### Resource availability

A list of critical reagents (key resources) is included in the Supplementary data. Relevant plasmids are available to the academic community. For additional materials, please email the lead contact for requests. Some material may require requests to collaborators and/or agreements with various entities.

### Reporting summary

Further information on research design is available in the Nature Portfolio Reporting Summary linked to this article.

## Data availability

The processed sequencing data in this paper have been deposited into the NCBI GEO database: GSE266329. Proteomics data have been deposited into PRIDE: PXD053872. Source Data are provided with this paper. All data supporting the findings of this study are included in the Supplementary Information or are available from the corresponding authors upon request. Raw numerical data underlying charts and graphs are provided in the Source Data file where applicable. Unique reagents used in this study are also available upon reasonable request. Source data are provided with this paper.

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

## Acknowledgements

We thank all members of the Feng lab as well as the DeSelm lab for helpful suggestions and technical advice. We thank USCF Laboratory for Cell Analysis core for cytometric and microscopic support and services. We thank WashU Mass Spec Core for lipidomic support and services. We thank UC Berkeley QB3 Genomics for sequencing services, RRID:SCR_022170. We thank 10x Genomics for single-cell RNA-Seq support and services. We thank Dr. Jeffrey Whitman for helpful advice on lipidomics. This study was supported by NIH grants 5DP5OD026427 to C.J.D.

## Author contributions

T.L., C.J.D. and F.Y.F. conceived, designed and initiated the study. T.F. assisted T.L. with various experiments. J.T.H. designed the CRISPR library. T.L. and H.L. performed CRISPR screens and sample pre-paration. M.Z. analyzed and visualized CRISPR screening results, proteomics and single-cell RNA-Sequencing data. D.Q. supervised all computational analysis in this study. A.W. supervised the proteomic experiment. A.K. and A.B. performed and analyzed proteomics data. S.G.L performed single-cell RNA-Seq sample preparation and helped with study design. J.Z. performed all animal experiments and histology experiments. H.J. and J.C. designed and engineered PBMC-derived HER-CAR-M. X.Z. analyzed patient data. A.B.K. generated α-EpHA2-CAR and α-CD19-CAR constructs. YA.G., J.K. and M.S. performed lipidomics and data analysis. A.S. and M.S. analyzed the RNA-sequencing results. K.C.F., J.S.C., L.A.G., and P.M.B. interpreted the data and provided suggestions for the CRISPR screen or data analysis. A.L. provided suggestions for the mechanistic study. T.L. and C.J.D. jointly prepared the manuscript with input from all authors. C.J.D. and F.Y.F. secured funding and supervised the work. All authors drafted or revised the article. All authors approved the final version of the manuscript.

## Competing interests

F.Y.F. has consulted for Astellas, Bayer, Blue Earth Diagnostics, BMS, EMD Serono, Exact Sciences, Foundation Medicine, Janssen Oncology, Myovant, Roivant, and Varian, and serves on the Scientific Advisory Board for BlueStar Genomics and SerImmune. F.Y.F. has patent applications with Decipher Biosciences, as well as with PFS Genomics/Exact Sciences in breast cancer, all unrelated to this work. L.A.G. has filed patents on CRISPR tools and CRISPR functional genomics, is a co-founder of Chroma Medicine, and a consultant for Chroma Medicine and Arena Bioworks. The remaining authors declare no competing interests.

## Additional information

[1]University of California, San Francisco, Helen Diller Family Comprehensive Cancer Center, San Francisco, CA 94158, USA. [2]Department of Radiation Oncology, University of California, San Francisco, San Francisco, CA 94158, USA. [3]Department of Pharmaceutical Chemistry, University of California, San Francisco, San Francisco, CA, USA. [4]Department of Laboratory Medicine, University of California, San Francisco, San Francisco, CA 94158, USA. [5]Gladstone–UCSF Institute of Genomic Immunology, San Francisco, CA 94158, USA. [6]Department of Medicine, University of California, San Francisco, San Francisco, CA 94143, USA. [7]Department of Radiation Oncology, Washington University School of Medicine, St. Louis, MO 63108, USA. [8]Bursky Center for Human Immunology and Immunotherapy, Washington University School of Medicine, St. Louis, MO 63110, USA. [9]Mass Spectrometry Technology Access Center at McDonnell Genome Institute (MTAC@MGI) at Washington University School of Medicine, St. Louis, MO 63108, USA. [10]Department of Clinical Sciences, Division of Oncology, Lund University, Lund, Sweden. [11]Division of Haematology, Oncology and Radiation Physics, Skåne University Hospital, Lund, Sweden. [12]Division of gynecologic oncology, University of California, San Francisco, CA 94158, USA. [13]Parker Institute for Cancer Immunotherapy, University of California, San Francisco, San Francisco, CA 94129, USA. [14]Department of Urology, University of California, San Francisco, CA 94158, USA. [15]Arc Institute, Palo Alto, CA 94304, USA. [16]Pharmaceutical Sciences and Pharmacogenomics, University of California San Francisco, San Francisco, CA 94143, USA. [17]Department of Ophthalmology, University of California San Francisco, San Francisco, CA 94115, USA. [18]Department of Epidemiology and Biostatistics, University of California, San Francisco, San Francisco, CA 94158, USA. [19]Department of Bioengineering and Therapeutic Sciences, University of California, San Francisco, San Francisco, CA 94143, USA. [20]Chan Zuckerberg Biohub San Francisco, San Francisco, CA 94158, USA. [21]These authors jointly supervised this work. ✉e-mail: deselmc@wustl.edu

