## [Transparent Peer Review file · Nature Communications]

Exploitable Mechanisms of Antibody and CAR Mediated Macrophage Cytotoxicity

Corresponding Author: Dr Carl DeSelm

Version 0:

Reviewer comments:

Reviewer #1

(Remarks to the Author)

Liu et al present a study on the role of cancer cell-expressed ATG9A in their resistance against macrophage cytotoxicity. The data are interesting, but the text too elaborate and not focused enough. I am not convinced that all supplemental data are equally helpful. The Introduction is too long and already provides an extended summary of the data, which is unnecessary. Specific comments are:

- 1) The conclusion of the first section is that ATG9A and UBAP1 are crucial regulators of resistance against macrophage cytotoxicity. However, in the pan-cancer and the ovarian cancer TCGA data (Fig S3B), ATG9A and UBAP1 predict opposite outcomes for the patient, which is to me unexpected in view of the data presented in this manuscript.
- 2) To evaluate whole cell phagocytosis, the authors measure double fluorescence. Double positivity in flow cytometry can also be due to the intense adhesion of two cells to each other. Microscopy should show actual uptake of the cancer cells.
- 3) The experiments shown in Fig 2F-G are no proof of an enhanced macrophage-mediated killing in the ATG9A-KO case. ADCC can also be mediated by other effector cells, such as NK cells. This should be proven by specifically eliminating macrophages or NK cells in vivo during Trastuzumab treatment.
- 4) Figure 3C. These data could be interpreted differently, as the macrophages are also exposed to a change in Ca²⁺. This may alter the cytotoxic mechanism of the macrophages, which is perhaps less influenced by ATG9A in the cancer cells.
- 5) If ROS is crucial in this whole mechanism, then ATG9A-KO should not have an effect on NK-mediated cytotoxicity, which is not ROS-mediated. This should be checked.
- 6) The data in Fig S6 are rather anecdotal/preliminary and do not allow a solid conclusion.
- 7) It is important to know to what extent ATG9A-KO is different from the KO of other autophagocytosis-related genes, when it comes to cancer cell survival. Are these differences significant? If not, the effect of ATG9A on autophagy can be considered as the most important survival mechanism, while the effect on membrane integrity would then be secondary.
- 8) Control tumors are bigger than ATG9A-KO tumors, which could be part of the explanation for the difference in macrophage composition. Test this on similarly sized tumors.
- 9) In addition, what is the role of the inflammatory cytokines in vivo?
- 10) Fig 6B. Show the feature plots for the characteristic genes of each cluster.
- 11) CSF1R inhibition could have been expected to have a bigger effect on those tumors that contain the largest CSF1R-hi population. Why is there no effect on control tumors?

Reviewer #2

(Remarks to the Author)

Liu et al identifies ATG9A as a key player in cancer cells' response to macrophage-mediated attacks, showing that its depletion enhances macrophage cytotoxicity. By inhibiting ATG9A-driven membrane repair and depleting non-cytotoxic macrophages, they offer a novel strategy to boost the effectiveness of CAR-macrophage or antibody therapies. There are several points to consider below:

Major Comments:

1. In the experiment examining ATG9A knockout's effect on cancer cell treatment sensitivity in vivo (Figures 2E-G), the source of macrophages is not identified. Mouse macrophages should be depleted and replaced with human macrophages to support their conclusions.

2. In the in vitro experiments (Figures 2A-D), the study examines the increased macrophage cytotoxicity following ATG9A knockout by co-culturing CAR-M cells with tumor cells. However, without conducting in vitro experiments to assess macrophage cytotoxicity after ATG9A knockout combined with trastuzumab, the in vivo experiments fail to compare the effects of trastuzumab treatment with CAR-M cell therapy. Simply replacing CAR-M cells with trastuzumab in the in vivo experiments (Figures 2E-G, 6G) is problematic, as trastuzumab does not exclusively recruit macrophages; other immune cells, such as NK cells, may also be involved. Therefore, the authors must include a CAR-M group in the in vivo experiments to confirm that ATG9A knockout specifically enhances macrophage cytotoxicity against tumor cells.
3. Additional experiments are necessary to further elucidate membrane damage and repair. For instance, while propidium iodide (PI) staining is used to indicate membrane damage (Figure 2C), further supporting evidence is required to thoroughly demonstrate both membrane damage and its subsequent repair.
4. In the experiment investigating the effect of ATG9A knockout on cancer cell sensitivity within a 3D co-culture system, the study controlled for the effect of sgRNA on tumor cell proliferation but did not evaluate its impact on macrophage proliferation. In Figure S2K, sgATG9A appears to enhance macrophage proliferation, suggesting that the observed increase in cytotoxicity may result from a higher number of macrophages rather than changes in tumor cell sensitivity to macrophage cytotoxicity.
5. In the experiment assessing the effect of ATG9A knockout on cancer cell membrane repair, the negative control group (sgGAL4) shows a continuous increase in LAMP1 membrane translocation, while the positive group (sgATG9A) exhibits a slight dip at 0.5 hours before returning to baseline. The sustained rise in LAMP1 levels in the negative control, rather than leveling off, may suggest ongoing lysosomal translocation required for membrane repair. If this trend were observed over a longer period, LAMP1 levels might peak and stabilize as the repair process concludes. However, if LAMP1 translocation continues indefinitely, it could lead to lysosome depletion, potentially disrupting cellular functions or even leading to cell death.
6. In the experiment examining the effect of ATG9A knockout on cancer cell survival under autophagy inhibition (Figure 5D), comparing survival rates between cells treated with and without ATG9A knockout alone does not fully capture the role of autophagy. A comparative analysis between ATG9A knockout cells with and without inhibitor treatment is needed to better understand how autophagy inhibition influences cell survival.
7. In the experiments evaluating ATG9A knockout's effect on cancer treatment sensitivity in vivo (Figures 2E-G) and the immunohistochemical analysis following combined treatment with a CSF1R inhibitor and trastuzumab (Figures 6H-J), the results for the trastuzumab treatment group are presented. However, in the experiment assessing the impact of ATG9A knockout tumors on the combination of a CSF1R inhibitor and trastuzumab (Figure 6G), a trastuzumab monotherapy group should be included as an additional control. This would enhance the experimental design and more clearly demonstrate the benefit of combination therapy over monotherapy.

Minor Comments:

1. In the experiment validating whether ATG9A regulates phagocytosis through these checkpoints (Figure S2J), alongside examining the checkpoints, it is essential to assess changes in the engulfment index using flow cytometry or fluorescence imaging. This approach would aid in excluding potential interference with the phagocytic process itself.
2. In the experiment exploring the synergistic role of ATG9A and CAVIN2 in membrane repair of cancer cells (Figure 3K), the y-axis label is missing in the figure.

Reviewer #3

(Remarks to the Author)

This manuscript from Liu et al. from Felix Feng's lab at UCSF and Carl DeSelm's group at WUSTL identifies ATG9A as a key mediator of tumor cell survival in the context of macrophage coculture. The manuscript represents a tremendous amount of work, although the primary claims of the paper are not necessarily strongly supported by the myriad experiments. In general, the experimental design is sound and the experiments are well-executed; the main reason the claims are not strongly supported is because many of the claims are around macrophage-mediated immunotherapy for cancer, whereas the vast majority of the interventions undertaken in the manuscript occur in a few tumor cell lines under investigation. Thus, it is not clear how "exploitable" these findings are.

The major weakness of the paper is in its model systems: much of the manuscript relies on THP-1 cells, which are a human monocytic AML line that can be made to differentiate into macrophages by treating them with phorbol esters (which the investigators do). They are then transduced with a CAR containing an Ag-binding domain coupled to the TM and cytoplasmic domains of FcGR1IA. These CAR-Ms are used to kill cancer cell lines OVCAR-8, SKOV-3, and Raji. Only Figure 2D-G and S2E were done with primary PBMC-derived macrophages; however, the macrophages in Figure 2 bear a HER2-CAR that has a CD3z cytoplasmic tail. It is not clear whether the S2E CAR-Ms express Eph2A- or HER2- binding CARs. Figure 6 was done with endogenous murine macrophages in vivo, treating SKOV3 cells. No tumors were used that were not cell lines (Figures S3 and S4 analyze clinical data with respect to ATG9A expression, but this obviously does not get at mechanism). Because most of the manuscript relies on cancer cell lines responding to THP-1 cells triggered by FcGR1IA stimulation, it is unclear how generalizable these survival mechanisms are. To support more fully the claims made in the text would be quite laborious, so it may be best to rewrite the text to support more modest claims.

Briefly, the authors show in vitro and in vivo that ATG9A depletion in cancer cell lines sensitizes them to macrophage-mediated killing. These experiments are convincing, although ATG9A depletion also generally seems to impair cancer cell growth in culture (although not so much in NSG tumor models). Proteomic and lipidomic analyses reveal that ATG9A deficiency impairs the cancer cell response to macrophage-induced plasma membrane damage through defective lysosomal exocytosis, reduced ceramide recruitment or production, and disrupted caveolar endocytosis; these experiments are the most meticulous and are largely convincing. The authors also claim that depleting CSF1R+ macrophages enhances the eradication of ATG9A-negative tumor cells; while it is true that CSF1R inhibition coupled with ATG9A-deletion leads to

less tumor growth than ATG9A knockout alone, it is not clear that this is a synergistic effect rather than an additive one.

Major concern: the authors contend that this work "elucidates how macrophages kill tumor cells, how tumor cells resist killing, and how one might improve tumor killing." This is a gross overstatement. In order to justify this claim, the authors would need to repeat Figures 2B and C with primary macrophages, and repeat the in vivo studies (2E-G and 6) with more physiologically-relevant tumor models - ideally, this would be done in a murine syngeneic/immunocompetent system with murine tumors; failing that, a human PDX model system could be used in NSG mice.

Minor concerns: most of this work is really about how ATG9A absence affects the ability of OVCAR8 and SKOV3 cells to combat plasma membrane damage caused by activated macrophages. The macrophages need to be activated, but they don't need to be CAR macrophages (and the inclusion of CARs with two different signaling domains - CD3z and FcGR1IA - is distracting and counterproductive). It may make sense to omit these experiments or to move them to supplemental, and focus on FcGR-mediated macrophage activation. This will also permit more work with primary macrophages rather than THP-1 cells, which would be helpful.

There are some missing controls, most notably in Figure 6 - because trastuzumab is already so effective at controlling SKOV3 tumors (Fig 2E), and because CSF1R inhibition is already very-well described as an intervention that improves anti tumor macrophage response, it is not surprising that adding CSF1R inhibition to trastuzumab improves tumor control. The claim of Figure 6, as written, is that "the combination of Trastuzumab and CSF1R inhibition to deplete non-cytotoxic macrophages significantly enhances ATG9A KO cancer cell killing in vivo." The word "enhances" suggests that the effect of CSF1R is somehow greater in ATG9A KO than not; this may be true, but if so needs to be supported with more rigorous experimentation. At a minimum, animals bearing tumors and treated with CSF1R but not trastuzumab should be included; even with this control, establishing synergy between CSF1R inhibition and ATG9A KO may not be possible - in that case, just changing the wording would suffice (although the main finding of the figure would then shift to H-J).

Reviewer #4

(Remarks to the Author)

Version 1:

Reviewer comments:

Reviewer #1

(Remarks to the Author)

The authors sufficiently addressed my concerns

Reviewer #2

(Remarks to the Author)

My concerns have been addressed in the revision and I recommend to accept it.

Reviewer #3

(Remarks to the Author)

Thank you for submitting this revised and improved manuscript. In general, it is much better and will be an important contribution to the field. As previously stated, my primary complaints were not about the science; rather, they were that the interpretation of the conclusions was overblown to the point of inaccuracy. While the revised manuscript is better, there are still significant overstatements that should be addressed, in particular with regards to the therapeutic potential/mechanism of these approaches. I would strongly urge the authors to revise the entire text, with particular attention to the abstract, introduction, and discussion, to align them with the data provided.

As a specific example, the authors write that "trastuzumab alone resulted in minimal tumor cell killing in the absence of macrophages," which is an over characterization of Figure S5G (which shows that cell death with trastuzumab alone goes from about 70% to about 45%, whereas trastuzumab + clo goes from about 45% to about 20%). Similarly, the authors write that "clodronate blunted the therapeutic effect of trastuzumab;" it would be far more accurate to say that clodronate blunted the effect of ATG9a knockout, as trastuzumab alone still has a significant effect after clo treatment.

Reviewer #4

(Remarks to the Author)

I co-reviewed this manuscript with one of the reviewers who provided the listed reports. This is part of the Nature Communications initiative to facilitate training in peer review and to provide appropriate recognition for Early Career

Researchers who co-review manuscripts.

REVIEWER COMMENTS

First, we thank the reviewers for the thoughtful and astute observations. We have taken every point seriously, performing substantial additional *in vitro* as well as *in vivo* experiments, which are now incorporated into the results. These additional findings significantly enhance the story, which we appreciate. Specific comments are addressed individually below.

Reviewer #1 (Remarks to the Author):

Liu et al present a study on the role of cancer cell-expressed ATG9A in their resistance against macrophage cytotoxicity. The data are interesting, but the text too elaborate and not focused enough. I am not convinced that all supplemental data are equally helpful. The Introduction is too long and already provides an extended summary of the data, which is unnecessary.

This is a helpful suggestion - the introduction has been shortened, particularly removing the summary of data, and the text has been edited to be less wordy and more focused.

Specific comments are:

1) The conclusion of the first section is that ATG9A and UBAP1 are crucial regulators of resistance against macrophage cytotoxicity. However, in the pan-cancer and the ovarian cancer TCGA data (Fig S3B), ATG9A and UBAP1 predict opposite outcomes for the patient, which is to me unexpected in view of the data presented in this manuscript.

Thank you for your observation, which brings up an interesting point regarding the potential roles of ATG9A and UBAP1 beyond what we describe here. This observation is one clue that influenced our decision to mainly focus our studies on understanding the role of ATG9A; low ATG9A levels are strongly correlated with better cancer survival by TCGA, consistent with our mouse findings and our mechanistic studies demonstrating ATG9A's critical role in protecting cancer cells from macrophage cytotoxicity. The observation that UBAP1, which we also identify as a regulator of cancer cell response to macrophage killing, actually has a positive correlation with cancer survival by TCGA (higher levels are better), likely point to the fact that UBAP1 has multiple other context-specific roles, and in the course of a patient's cancer treatment and progression, is not only involved in cancer cells' response to macrophage mediated cellular damage. UBAP1's involvement in ubiquitin-related pathways for example can influence the cellular stress responses, potentially leading to

enhanced cancer cell death under other therapeutic conditions. This points to ATG9A likely being a better therapeutic target for combining with macrophage targeted therapy broadly in cancer, whereas targeting UBAP1 would need to be more carefully considered in terms of specific context.

2) To evaluate whole cell phagocytosis, the authors measure double fluorescence. Double positivity in flow cytometry can also be due to the intense adhesion of two cells to each other. Microscopy should show actual uptake of the cancer cells.

We agree that target cell phagocytosis is a difficult phenomenon to definitively quantify by flow, as flow cytometry cannot accurately distinguish cell phagocytosis from intense adhesion between two cells, even with strict exclusion of doublets using both FSC-H vs. FSC-A and SSC-H vs. SSC-A plots as we have done.

To further validate with microscopy as suggested, we incorporated a live cell imaging-based approach using a new pHrodo-based phagocytosis assay to directly visualize cancer cell uptake over time. pHrodo Red is a dye that fluoresces only in acidic environments, thereby specifically indicating engulfment if the dye is transferred from the target cell to the CAR-M's acidic compartment (phagosome/lysosomes). We labeled sgGAL4 and sgATG9A OVCAR-8 tumor cells with pHrodo Red, co-cultured them with CAR-Ms in the IncuCyte system, and quantified red fluorescence signals over time within the CAR-Ms. Data were normalized to the 0-hour baseline and are presented in Figure S5A. As shown in Figure S5B, tumor phagocytosis events were observed but occurred at a low and equivalent frequency in both control and KO co-culture.

We hope this additional data and explanation address your concern.

3) The experiments shown in Fig 2F-G are no proof of an enhanced macrophage-mediated killing in the ATG9A-KO case. ADCC can also be mediated by other effector cells, such as NK cells. This should be proven by specifically eliminating macrophages or NK cells in vivo during Trastuzumab treatment.

Thank you for raising this important and critical concept. One initial important point is that these studies are in NSG mice, which we have now more clearly indicated in the figure legend. One benefit of NSG mice is that they are genetically devoid of NK cells (due to their IL2R γ null mutation, which renders them unresponsive to IL-2 receptor cytokines including IL-15; IL-15 signaling is essential for the development, survival, and function of NK cells). An immunological value of NSG mice is that the only immune cell line not genetically deleted is monocyte/macrophages. To demonstrate this is also the case in our NSG mice, in new Fig. S9A-B (pasted below), we show an analysis of sorted CD45⁺ mouse immune cells from

our NSG mice, demonstrating that the entire immune compartment expresses the monocyte/macrophage marker CD14, while the NK cell marker Ncr1 is basically absent:

However, to fully address your concern with experimental rigor, we did perform additional *in vivo* experiments using macrophage depletion in the context of antibody therapy in NSG mice, as well as *in vivo* macrophage reconstitution in the context of CAR-Ms, to definitively demonstrate we are observing macrophage-mediated tumor killing. ATG9A KO tumors in NSG mice pretreated with clodronate, which we confirmed substantially depleted the endogenous macrophage compartment, no longer significantly responded to Trastuzumab (Fig. 2G). Consistently, NSG mice pretreated with clodronate, followed by the injection of tumor cells and CAR-M treatment beginning on day 7, showed significant CAR-M mediated ATG9A KO tumor reduction relative to control tumors treated with CAR-Ms and relative to ATG9A KO tumors without CAR-Ms (Fig. 2I), demonstrating macrophage-specific killing of ATG9A-KO tumors *in vivo*.

We hope these additional data and explanations address your concern and strengthen the evidence for macrophage-mediated killing.

4) Figure 3C. These data could be interpreted differently, as the macrophages are also exposed to a change in Ca²⁺. This may alter the cytotoxic mechanism of the macrophages, which is perhaps less influenced by ATG9A in the cancer cells.

Thank you for this insightful comment. To address your concern, we conducted additional experiments to test whether our experimental conditions containing calcium or directly impact calcium function (Figure S6A; pasted below). We performed a 2-hour co-culture (the same experimental setup as in Figure 3C), but using varying effector-to-target (E:T) ratios of our CAR-Ms and WT tumor cells in media with or without Ca²⁺.

We observed that plasma membrane (PM) damage, as indicated by PI signal, increased with higher E:T ratios at identical rates in the presence or absence of Ca²⁺ in the media. These results demonstrate that macrophage cytotoxic potential is not influenced by the Ca²⁺ levels present in our experimental conditions.

5) If ROS is crucial in this whole mechanism, then ATG9A-KO should not have an effect on NK-mediated cytotoxicity, which is not ROS-mediated. This should be checked.

Thank you for your suggestion. We agree that based on published reports that NK cells (as well as T-cells) kill primarily by granzyme/perforin and sometimes FASL, TRAIL, or cytokines, ATG9A-KO should not have an effect on NK cell (or T-cell) cytotoxicity, assuming ATG9A has no additional role in susceptibility to granzyme/perforin/FASL/TRAIL or cytotoxic cytokines. While this is an interesting point, investigating the effect of ATG9A-KO on NK-mediated cytotoxicity lies outside the scope of this study. Our work focuses specifically on macrophage-mediated cytotoxic mechanisms and the role of ATG9A in this context. We hope this clarification is acceptable.

6) The data in Fig S6 are rather anecdotal/preliminary and do not allow a solid conclusion.

Thank you for your feedback. We agree that Figure S6 merely provides supporting evidence rather than a definitive conclusion. We included the data as a supplemental figure for this reason, as we believe it adds value to the study and provides potentially useful data for those actively researching the genes we genetically manipulated and assayed in this figure (IQGAP, CAVIN1, CAVIN2, UBAP1, CHMP6, CALR), without overextending the interpretation.

7) It is important to know to what extent ATG9A-KO is different from the KO of other autophagocytosis-related genes, when it comes to cancer cell survival. Are these differences significant? If not, the effect of ATG9A on autophagy can be considered as the most important survival mechanism, while the effect on membrane integrity would then be secondary.

Thank you for this thoughtful comment. To address this point, we further defined cancer cell survival following the knockout of individual autophagy-related genes, which is now included in Figure 5H (also pasted below). Our results show that knocking out these genes does not significantly affect tumor cell numbers on their own, and knocking out any of the other autophagy genes, except for ATG9A, does not affect tumor cell survival when cocultured with CAR-Ms. ATG9A seems to uniquely sensitize tumors to poor survival after coculture with CAR-Ms, and also seems to uniquely render the tumor cells immediately susceptible to membrane damage by CAR-Ms, as demonstrated by PI staining after short term (2hr) coculture.

These findings highlight that ATG9A KO is different from the KO of other autophagy-related genes, and that the role of ATG9A in maintaining membrane integrity when in coculture with targeted macrophages is more substantial than the overall survival mechanism of the autophagy pathway, at least in this nutrient replete setting (starvation may induce other effects, which may or may not be relevant to the typical cancer patient, and would fall in the scope for another study).

We hope these additional data and explanation address your concern.

8) Control tumors are bigger than ATG9A-KO tumors, which could be part of the explanation for the difference in macrophage composition. Test this on similarly sized tumors.

Thank you for your comment. We agree that tumor size could impact macrophage composition. It is important to note that the control tumors are not bigger than ATG9A-KO tumors at baseline, but only after treatment with antibody or CAR-Ms. The observation that ATG9A-KO tumors are much smaller than control tumors after therapy is a key finding of this study, as the substantial size reduction of ATG9A-KO tumors under Trastuzumab treatment demonstrates their increased sensitivity to this therapy. It is difficult to test macrophage composition on similarly sized tumors after Trastuzumab treatment because we find few small control tumors after Trastuzumab treatment, and even fewer large ATG9A KO tumors after Trastuzumab treatment.

However, to address your question, we selectively matched and compared sgGAL4 and sgATG9A tumors that were of most comparable size at the end of the experiment without Trastuzumab treatment, and quantified the macrophage number per square mm. These data show that ATG9A-KO tumors still exhibit a higher proportion of F4/80⁺ macrophages, suggesting that the observed difference in macrophage composition is independent of tumor size.

We hope this explanation and additional evidence address your concern.

9) In addition, what is the role of the inflammatory cytokines *in vivo*?

Thank you for this excellent question. As shown in Fig. S8 and Fig. 6, our data indicate that ATG9A KO induces the expression of numerous inflammatory cytokines by the tumor, and consistently *in vivo* there are not only more macrophages in these tumors, but the macrophages are more polarized towards an inflammatory phenotype. We would love to know the relative contribution of the array of induced inflammatory cytokines on this *in vivo* phenotype, but it is a difficult question to answer without extensive genetic knockout of the individual induced cytokines or their receptors. In the absence of these complicated knockout studies, our data do demonstrate that the inflammatory environment induced by ATG9A-KO in the tumor cells themselves is, as would be hypothesized, associated with enhanced recruitment and activation of cytotoxic macrophages; this may further contribute to the increased killing of ATG9A-KO tumors observed *in vivo*.

10) Fig 6B. Show the feature plots for the characteristic genes of each cluster.

This has been included in Figure S9.

11) CSF1R inhibition could have been expected to have a bigger effect on those tumors that contain the largest CSF1R-hi population. Why is there no effect on control tumors?

Thank you for this thoughtful question. You are correct that CSF1R inhibition might be expected to have a more pronounced effect on tumors with a high CSF1R⁺ population, if all else in the tumor were equal. The problem is that not all else is equal – the CSFR-hi control tumors have retained the capacity to effectively repair their damaged membrane after macrophage-mediated damage, so even though the tumor-supportive, CSF1R-hi macrophages have been significantly reduced, the tumor still has substantial defense against the CSF1R-low cytotoxic macrophages tethered to them by Trastuzumab. In contrast, the ATG9A-KO tumors have both lost their ability to repair their membranes damaged by the cytotoxic macrophages, and have lost the supportive CSF1R-hi macrophage population, so they are dramatically reduced.

In control tumors, CSF1Ri treatment did significantly slow tumor growth (Fig. 6I, comparing Trastuzumab sgGAL4 vs Trastuzumab + CSFRi sgGAL4). These data suggest that CSF1Ri treatment does have some effect on control tumors, although the impact is much more modest compared to ATG9A-KO tumors, due to their difference in membrane repair capacity.

We hope this clarification helps address your concern.

Reviewer #2 (Remarks to the Author):

Liu et al identifies ATG9A as a key player in cancer cells' response to macrophage-mediated attacks, showing that its depletion enhances macrophage cytotoxicity. By inhibiting ATG9A-driven membrane repair and depleting non-cytotoxic macrophages, they offer a novel strategy to boost the effectiveness of CAR-macrophage or antibody therapies. There are several points to consider below:

Major Comments:

1. In the experiment examining ATG9A knockout's effect on cancer cell treatment sensitivity in vivo (Figures 2E-G), the source of macrophages is not identified. Mouse macrophages should be depleted and replaced with human macrophages to support their conclusions. CAR-M?

We thank the reviewer for highlighting the importance of clarifying the source of macrophages used in this experiment. To address this, NSG mice (which we show possess no other immune cells besides monocyte/macrophages – see Figure S9A-B) were pretreated with clodronate liposomes to deplete endogenous macrophages before the injection of SKOV3-LUC tumor cells. Subsequently, five million EphA2-CAR macrophages were injected intraperitoneally into control or ATG9A-KO tumor bearing mice (Fig. 2I).

Tumor sizes were measured at Day 28, and the data demonstrated significantly smaller tumors in the ATG9A-KO group compared to the control group following CAR-M treatment, as well as compared to the ATG9A-KO tumor without CAR-M treatment. These revised experiments clarify the direct role of macrophages in ATG9A-deficient tumor cell sensitivity. See comment #3 above for further discussion of this important topic.

2. In the in vitro experiments (Figures 2A-D), the study examines the increased macrophage cytotoxicity following ATG9A knockout by co-culturing CAR-M cells with tumor cells. However, without conducting in vitro experiments to assess macrophage cytotoxicity after ATG9A knockout combined with trastuzumab, the in vivo experiments fail to compare the effects of trastuzumab treatment with CAR-M cell therapy. Simply replacing CAR-M cells with trastuzumab in the in vivo experiments (Figures 2E-G, 6G) is problematic, as trastuzumab does not exclusively recruit macrophages; other immune cells, such as NK cells, may also be involved. Therefore, the authors must include a CAR-M group in the in vivo experiments to confirm that ATG9A knockout specifically enhances macrophage cytotoxicity against tumor cells.

We thank the reviewer for this insightful comment, which has prompted us to clarify our approach and perform additional in vivo experiments to further strengthen the manuscript. Please see comment #3 above as well.

1. Absence of NK Cells in the NSG Model:

To address the concern about NK cell involvement, we analyzed sorted mouse immune cells in the NSG model and confirmed, as is expected from the genetic absence of NK cells in NSG mice, that the entire immune component in our NSG mice express the monocyte/macrophage marker CD14. In contrast, the NK cell population (Ncr1+) was essentially absent. This indicates that the antibody-dependent cellular cytotoxicity (ADCC) observed in this model is mediated exclusively by macrophages, as NK cells are effectively absent.

2. In Vitro Co-Culture Results:

To further assess macrophage cytotoxicity following ATG9A knockout in combination with trastuzumab, we conducted in vitro co-culture experiments, presented in Figure S5. The results demonstrate that trastuzumab alone produced only modest tumor cell killing in the absence of macrophages. However, in a co-culture setting with macrophages, trastuzumab significantly enhanced tumor cell killing in sgGAL4 tumor and even more so in sgATG9A tumor (S5G).

3. **Additional In Vivo Experiment with CAR-Ms:**

To further address the reviewer's request and confirm that ATG9A knockout specifically enhances macrophage cytotoxicity in vivo using depletion and reconstitution studies, we performed an additional in vivo experiment using the CAR-M model, as described in our response to the previous comment. These data reinforce the conclusion that ATG9A knockout significantly enhances macrophage-mediated tumor killing.

Revisions reflecting these points have been incorporated into the manuscript, including updated figures and descriptions in the Results and Methods sections. We believe these additions significantly improve the robustness of the conclusions and provide a more comprehensive analysis of the roles of CAR-Ms and trastuzumab in this context; thank you for these comments.

3. Additional experiments are necessary to further elucidate membrane damage and repair. For instance, while propidium iodide (PI) staining is used to indicate membrane damage (Figure 2C), further supporting evidence is required to thoroughly demonstrate both membrane damage and its subsequent repair.

We thank the reviewer for this valuable suggestion, which has helped us further strengthen the manuscript. To address this, we added an additional experiment using a HaloTag probe, which utilized both membrane permeable ligand and impermeable ligand to detect early-stage plasma membrane (PM) damage. The HaloTag system provides a dynamic and mechanistic understanding of PM disruptions, allowing us to visualize subtle changes in PM integrity at an earlier stage compared to PI staining. This data, which was incorporated into Fig. 3D-E, demonstrate that ATG9A deficiency does not result in any difference in membrane permeability at baseline, but rapidly after coculture with CAR macrophages (two hours, before cell death occurs), the membrane impermeable lipid significantly accumulates in the ATG9A deficient tumor cells.

Furthermore, results from our membrane lipidomics data quantified PM damage at the lipid level, showing that macrophage nibbling and ROS-induced lipid peroxidation are the primary factors driving PM damage and initiating repair processes. While Figure 3 focuses on the mechanisms underlying PM damage, Figure 4 presents a comprehensive analysis of the repair process, including lysosome exocytosis, ceramide enrichment, and caveolar endocytosis. We also show that providing additional recombinant ASMase, which mediates membrane repair, and we find is deficient in ATG9A-KO tumor cells, brings intracellular PI accumulation completely back to baseline in ATG9A-KO tumor cells cocultured with macrophages (Fig. 4I), demonstrating the definitive link between ATG9A KO and membrane repair after macrophage coculture. We believe these additional experiments and analyses provide a thorough and mechanistic understanding of PM damage and repair, and we thank you for these suggestions.

4. In the experiment investigating the effect of ATG9A knockout on cancer cell sensitivity within a 3D co-culture system, the study controlled for the effect of sgRNA on tumor cell proliferation but did not evaluate its impact on macrophage proliferation. In Figure S2K, sgATG9A appears to enhance macrophage proliferation, suggesting that the observed increase in cytotoxicity may result from a higher number of macrophages rather than changes in tumor cell sensitivity to macrophage cytotoxicity.

We thank the reviewer for this insightful observation. To address this concern, we have added Figure S5F, which includes IncuCyte quantification of macrophage GFP mean fluorescence intensity (MFI) over time in the 3D spheroid co-culture system. The data indicate that macrophages did not exhibit significant proliferation in either the sgGAL4 or sgATG9A spheroids. Furthermore, there was no observable difference in macrophage proliferation between the sgGAL4 and sgATG9A groups, suggesting that the enhanced cytotoxicity observed in the sgATG9A condition is not attributable to increased macrophage proliferation. Other additional experiments, such as those in Fig. 4 showing that restoring the membrane repair protein ASMase which is downregulated in ATG9A KO cells, equalizes

both the two hour membrane damage as well as the 3 day tumor cell survival between ATG9A KO and control tumor cells cocultured with macrophages, demonstrate this is due to a tumor intrinsic effect and not through an indirect effect on the macrophages. This additional analysis strengthens our conclusion that ATG9A knockout specifically enhances tumor cell sensitivity to macrophage-mediated cytotoxicity through membrane repair defects.

5. In the experiment assessing the effect of ATG9A knockout on cancer cell membrane repair, the negative control group (sgGAL4) shows a continuous increase in LAMP1 membrane translocation, while the positive group (sgATG9A) exhibits a slight dip at 0.5 hours before returning to baseline. The sustained rise in LAMP1 levels in the negative control, rather than leveling off, may suggest ongoing lysosomal translocation required for membrane repair. If this trend were observed over a longer period, LAMP1 levels might peak and stabilize as the repair process concludes. However, if LAMP1 translocation continues indefinitely, it could lead to lysosome depletion, potentially disrupting cellular functions or even leading to cell death.

We thank the reviewer for this detailed observation and thoughtful question. To further investigate this trend and distinguish these possibilities, we extended the monitoring period by including additional time points (every 30 minutes up to 5 hours) and observed that LAMP1 levels on the cancer cell membrane in the sgGAL4 co-culture gradually increased before reaching a plateau at around 4.5 hours (Figure 4E), while LAMP1 levels on sgATG9A cells remained to be low throughout the co-culture.

The plateau in LAMP1 levels suggests that the lysosomal translocation process reaches a steady state. This stabilization likely reflects a balance between lysosomal exocytosis and stabilization of PM repair signaling, preventing indefinite lysosomal fusion with the PM that could deplete cellular lysosome stores.

Overall, these observations highlight the dynamics of lysosomal translocation during PM repair and suggest that in cells with intact ATG9A, the repair process may be finely regulated to prevent lysosome depletion and maintain cellular homeostasis. These findings have been incorporated into the revised manuscript and we thank the reviewer for these insightful comments.

6. In the experiment examining the effect of ATG9A knockout on cancer cell survival under autophagy inhibition (Figure 5D), comparing survival rates between cells treated with and without ATG9A knockout alone does not fully capture the role of autophagy. A comparative analysis between ATG9A knockout cells with and without inhibitor treatment is needed to better understand how autophagy inhibition influences cell survival.

We sincerely appreciate the reviewer's thoughtful comment and valuable scientific insight. You've highlighted an important consideration regarding the role of autophagy in ATG9A-mediated sensitivity, which has helped us refine the interpretation of our data.

In Figure 5D, our goal was to determine whether ATG9A-mediated sensitivity to macrophage killing is linked to the autophagy pathway. To address this, we co-cultured a 1:1 mixture of control (sgGAL4) and ATG9A knockout (sgATG9A) cells with CAR-Ms. We observed two key findings: ATG9A knockout cells were consistently less abundant than control cells after co-culture; This effect was maintained even in the presence of the autophagy inhibitor Autophinib, suggesting that ATG9A-mediated sensitivity is not solely dependent on autophagy.

In line with your suggestion, we performed an additional analysis presented in Figure 5E to directly compare ATG9A knockout cells with and without Autophinib treatment. Interestingly, we found that Autophinib treatment did not further enhance macrophage-mediated killing of ATG9A knockout cells. This result supports the

conclusion that ATG9A-mediated sensitization operates through an autophagy-independent mechanism.

Our results also show that knocking out a variety of key autophagy genes does not significantly affect tumor survival on their own, and knocking out any of the other autophagy genes, except for ATG9A, has no effect on tumor cell survival when cocultured with CAR-Ms. ATG9A seems to uniquely sensitize tumors to poor survival after coculture with CAR-Ms, and also seems to uniquely render the tumor cells susceptible to membrane damage by CAR-Ms, as demonstrated by PI staining after short term (2hr) coculture (Fig. 5 F-H).

We are truly grateful for your suggestion, which allowed us to strengthen our data and clarify this key aspect of our study.

7. In the experiments evaluating ATG9A knockout's effect on cancer treatment sensitivity in vivo (Figures 2E-G) and the immunohistochemical analysis following combined treatment with a CSF1R inhibitor and trastuzumab (Figures 6H-J), the results for the trastuzumab treatment group are presented. However, in the experiment assessing the impact of ATG9A knockout tumors on the combination of a CSF1R inhibitor and trastuzumab (Figure 6G), a trastuzumab monotherapy group should be included as an additional control. This would enhance the experimental design and more clearly demonstrate the benefit of combination therapy over monotherapy.

We thank the reviewer for this thoughtful suggestion. While the results for the trastuzumab-only group were initially presented in Figure S2E, we agree that including this group directly in Figure 6 provides a clearer comparison. To address this, we have added the trastuzumab-only group in Figure 6I, allowing for a direct evaluation of the combination therapy's benefit over monotherapy.

Additionally, we included a CSF1Ri-only control group in Figure 6G, which demonstrates that CSF1Ri treatment alone does not dramatically impact tumor growth. Together, these additional comparisons strengthen the experimental conclusions and provide a more comprehensive analysis of the therapeutic effects of the combination treatment; thank you for the suggestion.

Minor Comments:

1. In the experiment validating whether ATG9A regulates phagocytosis through these checkpoints (Figure S2J), alongside examining the checkpoints, it is essential to assess changes in the engulfment index using flow cytometry or fluorescence imaging. This approach would aid in excluding potential interference with the phagocytic process itself.

We thank the reviewer for this valuable suggestion. To address this, we have more thoroughly assessed macrophage phagocytosis of cancer cells as presented in Figure 2C and Figure S5A-B. These new analyses include measurements of the engulfment index using flow cytometry-based method as well as a PH-sensitive dye-based microscopy method, which confirms that ATG9A knockout does not interfere with the phagocytic process itself (for more details on this particular experiment, see response to Reviewer 1's comment #2 above).

2. In the experiment exploring the synergistic role of ATG9A and CAVIN2 in membrane repair of cancer cells (Figure 3K), the y-axis label is missing in the figure.

We appreciated your comment, we've changed the y-axis.

Reviewer #3 summary of comments:

This manuscript from Liu et al. from Felix Feng's lab at UCSF and Carl DeSelm's group at WUSTL identifies ATG9A as a key mediator of tumor cell survival in the context of macrophage coculture. The manuscript represents a tremendous amount of work, although the primary claims of the paper are not necessarily strongly supported by the myriad experiments. In general, the experimental design is sound and the experiments are well-executed; the main reason the claims are not strongly supported is because many of the claims are around macrophage-mediated immunotherapy for *cancer*, whereas the vast majority of the interventions undertaken in the manuscript occur in a few *tumor cell lines* under investigation.

Briefly, the authors show in vitro and in vivo that ATG9A depletion in cancer cell lines sensitizes them to macrophage-mediated killing. These experiments are convincing, although ATG9A depletion also generally seems to impair cancer cell growth in culture (although not so much in NSG tumor models). Proteomic and lipidomic analyses reveal that ATG9A deficiency impairs the cancer cell response to macrophage-induced plasma membrane damage through defective lysosomal exocytosis, reduced ceramide recruitment or production, and disrupted caveolar endocytosis; these experiments are the most meticulous and are largely convincing.

Thank you for the helpful assessment of strengths (above) and weaknesses (below); your raised concerns below are organized and addressed from major to minor:

The major weakness of the paper is in its model systems: much of the manuscript relies on THP-1 cells, which are a human monocytic AML line that can be made to differentiate into macrophages by treating them with phorbol esters (which the investigators do). They are then transduced with a CAR containing an Ag-binding domain coupled to the TM and cytoplasmic domains of FcγRIIA. These CAR-Ms are used to kill cancer cell lines OVCAR-8, SKOV-3, and Raji. Only Figure 2D-G and S2E were done with primary PBMC-derived macrophages; however, the macrophages in Figure 2 bear a HER2-CAR that has a CD3z cytoplasmic tail.

Thank you for bringing up this important weakness. We agree that observations in THP-1 cells far from guarantee the effects extend faithfully to primary macrophages. Thus, to first validate more of the key observed phenotypes in true macrophages, we have added more primary PBMC-derived macrophage results (Figure S5C shows repeated co-culture experiments using primary PBMC-derived macrophages rather than THP-1 cells; these experiments demonstrate that ATG9A knockout sensitizes cancer cells to PBMC-derived macrophage-induced plasma membrane damage rather than direct phagocytosis), and have used clodronate to deplete macrophages *in vivo* to demonstrate the effects extend to endogenous primary macrophages present in NSG mice. Second, we have used more specific language to not overextend THP-1 results. Specific language changes include:

We use “two distinct CAR-THP-1 cell lines: **liquid tumor relevant** CD19-targeting CAR-THP-1 (**Figure S1A-B**), and **solid tumor relevant** EphA2-targeting CAR-THP-1 (**Figure S1C-D**). Unless otherwise specified, these **CAR-THP-1 cells** will be referred to as α -EphA2-Ms and α -CD19-CAR-Ms, respectively throughout this paper.”

We also included this caveat in the discussion:

“To make feasible many of the large-scale studies performed here, many of our *in vitro* studies were done using THP-1 cells with confirmation of key results using primary human macrophages as well as endogenous mouse for *in vivo* studies. However, a caveat of this approach is that THP-1 cells may not approximate primary macrophages in every *in vitro* condition tested.”

Thus, we have extended experimental validation to include more studies with two types of primary macrophages (human primary macrophages and endogenous mouse macrophages) rather than just THP-1 cells, have further drawn attention to which experiments are THP-1 cells and pointed out the caveat of extending all THP-1 results to primary macrophages.

The general comments section (first paragraph) as well as the specific comments section (second paragraph) raise this additional valid concern:

Figure 6 was done with endogenous murine macrophages *in vivo*, treating SKOV3 cells. No tumors were used that were not cell lines (Figures S3 and S4 analyze clinical data with respect to ATG9A expression, but this obviously does not get at mechanism). Because most of the manuscript relies on *cancer cell lines* responding to THP-1 cells triggered by FcGR1IA stimulation, it is unclear how generalizable these survival mechanisms are. To

support more fully the claims made in the text would be quite laborious, so it may be best to rewrite the text to support more modest claims.

Major concern: the authors contend that this work "elucidates how macrophages kill tumor cells, how tumor cells resist killing, and how one might improve tumor killing." This is a gross overstatement. In order to justify this claim, the authors would need to repeat Figures 2B and C with primary macrophages, and repeat the *in vivo* studies (2E-G and 6) with more physiologically-relevant tumor models - ideally, this would be done in a murine syngeneic/immunocompetent system with murine tumors; failing that, a human PDX model system could be used in NSG mice.

Thank you for pointing out the overly broad cancer claims, in particular the claim that these results "elucidate how macrophages kill tumor cells, how tumor cells resist killing, and how one might improve tumor killing." We did two things in response to this –

First, we rewrote the text as suggested, to avoid or remove oversimplified and overextended summaries that, as you point out, would require an extensive amount of additional *in vivo* testing to justify.

Specific changes to the language include the following:

We removed this statement entirely: "This work elucidates how macrophages directly kill tumor cells when targeting them with a CAR or antibody, and how tumor cells resist macrophage killing. It also demonstrates novel target combinations to maximize tumor killing by cytotoxic macrophages, endowing them with the ability to achieve complete responses in mice even in the absence of any antitumor T-cell response."

We revised the original statement from: "This study showed ATG9A plays an important role in regulating cancer cell response to macrophage killing."

To: "ATG9A depletion sensitizes ovarian cancer cells to killing by CAR-THP-1 macrophages."

In the last paragraph of the discussion, we were careful to be more specific in the summary of results: "As summarized in **Figure 7**, our study reveals that ATG9A plays a multifaceted role in regulating SKOV3 cancer cell responses to macrophage-induced cytotoxicity..."

Additionally, we have clarified the experimental details where necessary, particularly in the SKOV3 mouse model sections.

To help clarify the cancer cell lines we used to demonstrate and validate results, we made this table to summarize key findings for our self and reviews:

Models	α -EphA2-CAR-THP-1-macrophages		α -CD19-CAR-THP-1-macrophages	PBMC macrophages	PBMC-derived HER2-CAR-macrophages
co-culture CRISPR screen	OVCAR-8		OVCAR-8		
	Figure 1C		Figure 1D		
Screen validation	OVCAR-8	SKOV3	OVCAR-8	OVCAR-8	
	Figure 1G	Figure S2C	Figure 1G	Figure S2E	
In vitro killing assay	OVCAR-8	SKOV3	OVCAR-8	OVCAR-8	SKOV3
	Figure 1G-I	Figure S2C	Figure 1G	Figure S5C	Figure S5D
In vitro PM integrity assay	OVCAR-8			OVCAR-8	SKOV3
	Figure 2C			Figure S5C	Figure S5D
In vivo subcutaneous SKOV3 tumor model	SKOV3 + endogenous mouse macrophages				
In vivo intraperitoneal SKOV3 tumor model	SKOV3 + α -EphA2-CAR-THP-1-macrophages				

We did also validate the *in vitro* killing assays and plasma membrane damaging assays using two additional cancer cell lines, LNCAP and 22RV1, with consistent results, however since these are prostate cancer cells we did not include them as we felt it added too much additional complexity.

Regarding additional *in vivo* mouse models, we did begin to establish a new ATG9A KO *in vivo* syngeneic tumor model to further attempt to extend the results. Specifically, we used a syngeneic BL6 mouse ovarian cancer cell line, ID8, which naturally expresses EphA2, and transduced it with GFP-Luciferase plus CRISPR/Cas9 and either control or ATG9A gRNA. We selected control and ATG9A KO clones, and grew them in B6 mice via IP injection, then treated both control and knockout tumor bearing mice with either nothing, control macrophages, or EphA2-CAR macrophages injected IP. What we unfortunately discovered is that the modifications we made to the tumor (most likely the Cas9/GFP/Luc) rendered it fairly immunogenic, as all conditions (including untreated tumors) rejected on their own, seemingly through a T-cell mediated mechanism.

We recognize that rather than making the broad claim that these results extend to “cancer” in general, which ideally would require a syngeneic mouse model or human PDX cancer models to be optimized and tested with this KO, the much more realistic approach is to tailor the conclusions to the models we have used. We believe the additional *in vivo* experiments in NSG mice in which we depleted macrophages with clodronate and abrogated the *in vivo* effect observed with antitumor antibody and ATG9A KO help demonstrate more conclusively the ATG9A KO SKOV3 tumor elimination phenotype observed is due to endogenous macrophages. Our caveats discussion was modified to specifically point out these results using SKOV3 cells *in vivo* and OVCAR8 and SKOV3 cells

in vitro may not extend to other cancer models, and also do not take into account the impact of other immune cells, etc:

“TCGA analysis shows lower ATG9A mRNA is associated with better survival on the pan-cancer level, as well as in a number of individual cancers such as ovarian cancer. We focused on ovarian cancer cell lines, with *in vivo* studies using SKOV3 ovarian cancer cells, and supporting *in vitro* data from both SKOV3 and OVCAR-8 ovarian cancer cells. Our studies did not use syngeneic tumors, primary human PDX models, or non-ovarian cancer models, but future studies in these models would help elucidate the impact of other immune cells in ATG9A deficient tumors, cancer heterogeneity, and ATG9A’s function in other tumor types.”

If the reviewer feels that these results really need to be replicated in an immune competent mouse model, we can further optimize and test this, however we hope that the removed statement above, as well as toned down and more specific language plus the included additional data suffice to address this concern. We are grateful for the reviewer’s insights, which have significantly strengthened the clarity and focus of our manuscript.

Minor concerns: most of this work is really about how ATG9A absence affects the ability of OVCAR8 and SKOV3 cells to combat plasma membrane damage caused by activated macrophages. The macrophages need to be activated, but they don't need to be CAR macrophages (and the inclusion of CARs with two different signaling domains - CD3z and FcGR1IA - is distracting and counterproductive). It may make sense to omit these experiments or to move them to supplemental and focus on FcGR-mediated macrophage activation. This will also permit more work with primary macrophages rather than THP-1 cells, which would be helpful.

We sincerely appreciate the reviewer’s thoughtful suggestion, which has helped us improve the clarity and focus of the manuscript. In response, we moved the CD3z HER2-CAR macrophage co-culture experiment to Figure S5D, and focused on FcGR-mediated macrophage activation as the mechanism in the main text. This adjustment eliminates potential distractions caused by the inclusion of CARs with different ITAM-based signaling domains and provides a more cohesive narrative.

These two comments (one from the general response, and the other from the specific concerns) address a similar topic, so are answered together:

The authors show *in vitro* and *in vivo* that ATG9A depletion in cancer cell lines sensitizes them to macrophage-mediated killing. These experiments are convincing, although ATG9A

depletion also generally seems to impair cancer cell growth in culture (although not so much in NSG tumor models). Proteomic and lipidomic analyses reveal that ATG9A deficiency impairs the cancer cell response to macrophage-induced plasma membrane damage through defective lysosomal exocytosis, reduced ceramide recruitment or production, and disrupted caveolar endocytosis; these experiments are the most meticulous and are largely convincing. The authors also claim that depleting CSF1R+ macrophages enhances the eradication of ATG9A-negative tumor cells; while it is true that CSF1R inhibition coupled with ATG9A-deletion leads to less tumor growth than ATG9A knockout alone, it is not clear that this is a synergistic effect rather than an additive one.

There are some missing controls, most notably in Figure 6 - because trastuzumab is already so effective at controlling SKOV3 tumors (Fig 2E), and because CSF1R inhibition is already very-well described as an intervention that improves anti-tumor macrophage response, it is not surprising that adding CSF1R inhibition to trastuzumab improves tumor control. The claim of Figure 6, as written, is that "the combination of Trastuzumab and CSF1R inhibition to deplete non-cytotoxic macrophages significantly enhances ATG9A KO cancer cell killing in vivo." The word "enhances" suggests that the effect of CSF1R is somehow greater in ATG9A KO than not; this may be true, but if so needs to be supported with more rigorous experimentation. At a minimum, animals bearing tumors and treated with CSF1R but not trastuzumab should be included; even with this control, establishing synergy between CSF1R inhibition and ATG9A KO may not be possible - in that case, just changing the wording would suffice (although the main finding of the figure would then shift to H-J).

This thoughtful feedback helped us improve the rigor and clarity of our study. In response to your suggestion, we included a CSF1R inhibitor (CSF1Ri) monotherapy group as an additional control and repeated the SKOV-3 tumor model to specifically assess the effect of CSF1Ri alone. The updated data are now presented in Figure 6.

Given the complexity of our *in vivo* experiments and the multiple treatment groups involved, we have modified Figure 6 to enhance clarity:

Figure 6G now shows the CSF1Ri-only groups, confirming that CSF1Ri monotherapy does not significantly affect tumor growth in either the sgGAL4 or sgATG9A groups. This result confirmed the absence of antibody-mediated macrophage cytotoxicity when Trastuzumab is not present.

Figure 6H presents the combination treatment group (CSF1Ri + Trastuzumab), allowing a direct comparison with monotherapies.

Figure 6I highlights the key comparison between the combination therapy and Trastuzumab monotherapy on control and ATG9A KO tumors. Our data demonstrate that the combination therapy provides an enhanced therapeutic benefit, particularly in the ATG9A KO group, where nearly all tumors completely regressed after treatment.

Establishing synergy between CSF1R inhibition and ATG9A knockout may be challenging, however these data show an enhanced therapeutic effect when these interventions are combined; neither CSF1Ri nor Trastuzumab alone are nearly as effective as when combined in ATG9A KO tumor bearing mice. We agree that more detailed analyses would be required to formally define synergy, and we have adjusted the wording in the manuscript to reflect this nuance, focusing on the enhanced therapeutic benefit rather than definitive synergy.

These additional experiments and adjustments strengthen our conclusions by clearly demonstrating the therapeutic advantage of combining CSF1R inhibition with Trastuzumab, especially in the context of ATG9A depletion. We are truly grateful for your constructive feedback, which has helped improve both the experimental design and the clarity of our manuscript.

It is not clear whether the S2E CAR-Ms express Eph2A- or HER2- binding CARs.

These are PBMC derived macrophages with no CAR. This has been clarified in the figure legend.

Reviewer #4 (Remarks to the Author):

We appreciate the opportunity to address the feedback provided by the reviewer, we also extend our gratitude to the Early Career Researcher who co-reviewed this manuscript, as their thoughtful comments have greatly contributed to improving the quality and clarity of our work. Thank you for recognizing and supporting this collaborative effort in peer review.

REVIEWERS' COMMENTS

Reviewer #1 (Remarks to the Author):

The authors sufficiently addressed my concerns

Reviewer #2 (Remarks to the Author):

My concerns have been addressed in the revision and I recommend to accept it.

Reviewer #3 (Remarks to the Author):

Thank you for submitting this revised and improved manuscript. In general, it is much better and will be an important contribution to the field. As previously stated, my primary complaints were not about the science; rather, they were that the interpretation of the conclusions was overblown to the point of inaccuracy. While the revised manuscript is better, there are still significant overstatements that should be addressed, in particular with regards to the therapeutic potential/mechanism of these approaches. I would strongly urge the authors to revise the entire text, with particular attention to the abstract, introduction, and discussion, to align them with the data provided.

As a specific example, the authors write that "trastuzumab alone resulted in minimal tumor cell killing in the absence of macrophages," which is an over characterization of Figure S5G (which shows that cell death with trastuzumab alone goes from about 70% to about 45%, whereas trastuzumab + clo goes from about 45% to about 20%). Similarly, the authors write that "clodronate blunted the therapeutic effect of trastuzumab;" it would be far more accurate to say that clodronate blunted the effect of ATG9a knockout, as trastuzumab alone still has a significant effect after clo treatment.

We thank the reviewer for the thoughtful and constructive feedback, and we greatly appreciate the recognition of the manuscript's scientific contribution. We recognize overstated aspects of the manuscript remain, and we have carefully revised the text throughout to better align our conclusions with the presented data.

In particular, we have:

- Addressed Reviewer #3's comment regarding the interpretation of **Figure S5G**, by revising the text to replace “minimal tumor killing” with “moderate tumor killing” and have clarified that “clodronate blunted the therapeutic effect of ATG9A knockout” rather than that of trastuzumab alone.
- We have revised the text to replace “robustly enhances tumor eradication in therapeutic antibody-treated mice” with “enhances the anti-tumor activity of therapeutic antibodies in NSG mice”
- We have revised the text to replace “underscoring the therapeutic potential of targeting ATG9A in cancer treatment” with “suggesting that targeting ATG9A may improve responses to macrophage-based therapies.”
- We have revised the text to replace “The disruption of these mechanisms in ATG9A knockout cells highlights potential therapeutic targets of membrane repair in cancer treatment” with “Disruption of these processes in ATG9A-deficient cells reveals potential vulnerabilities in membrane repair pathways that may be exploited for cancer therapy.”
- We revised the text to replace “highlighting ATG9A as a potential therapeutic target for macrophage-mediated cancer immunotherapy.” with “These findings support the potential of targeting ATG9A to improve the efficacy of macrophage-based cancer immunotherapies.”
- We have revised the text to remove “Finally, the therapeutic implications of our findings are underscored by our *in vivo* studies.”
- We have revised the text to replace “This dramatic tumor regression underscores the potential of this combined therapeutic strategy.” with more descriptive language “Depleting the macrophage population with clodronate abrogated this effect.”
- We have revised the text to replace “can significantly enhance therapeutic outcomes in ATG9A-deficient tumors.” with “contribute to enhanced control of ATG9A-deficient tumors.”
- We have revised the text to replace “Additionally, the significant tumor regression observed with the combination of Trastuzumab and CSF1R inhibition underscores the therapeutic potential of targeting both cancer cells and the myeloid compartment to enhance macrophage cytotoxic activity and improve overall treatment outcomes.” with “Additionally, the tumor regression observed with the combination of trastuzumab and CSF1R inhibition suggests that simultaneously

targeting cancer cells and modulating the myeloid compartment may enhance macrophage-mediated cytotoxicity.”

- We have revised the text to replace “to further **develop ATG9A-targeted therapies** to effectively harness macrophage-based cytotoxicity,” with “for further **investigation of ATG9A-targeted approaches** as a means to augment macrophage-based therapies,”
- Revised the **abstract, introduction, and discussion** to moderate the language around therapeutic potential, making it clear that our findings support **potential strategies** and **observed enhancements**, rather than definitive or broadly generalizable conclusions.

We believe these revisions improve the clarity and accuracy of the manuscript and better reflect the scope of the data. We thank the reviewer again for helping us improve the rigor and presentation of our work.

Reviewer #4 (Remarks to the Author):
